# Uniformly elevated future heat stress in China driven by spatially heterogeneous water vapor changes

Fan Wang[1,2,6], Meng Gao [1,2,6] ✉, Cheng Liu [3,4] ✉, Ran Zhao[4,5] & Michael B. McElroy [2]

The wet bulb temperature ($T_w$) has gained considerable attention as a crucial indicator of heat-related health risks. Here we report south-to-north spatially heterogeneous trends of $T_w$ in China over 1979-2018. We find that actual water vapor pressure ($E_a$) changes play a dominant role in determining the different trend of $T_w$ in southern and northern China, which is attributed to the faster warming of high-latitude regions of East Asia as a response to climate change. This warming effect regulates large-scale atmospheric features and leads to extended impacts of the South Asia high (SAH) and the western Pacific subtropical high (WPSH) over southern China and to suppressed moisture transport. Attribution analysis using climate model simulations confirms these findings. We further find that the entire eastern China, that accommodates 94% of the country's population, is likely to experience widespread and uniform elevated thermal stress the end of this century. Our findings highlight the necessity for development of adaptation measures in eastern China to avoid adverse impacts of heat stress, suggesting similar implications for other regions as well.

Unprecedented heat extremes have been ravaging the globe in recent years[1–4]. July 2023 was confirmed as the hottest month on record, ~1.5 °C warmer than the pre-industrial level[5]. Elevated heat stress has emerged as a prominent global climate concern and the upward trend is expected to intensify due to ongoing global warming[6]. The impact of heat stress is mediated by moisture levels, and metrics are critical for risk assessment of moist heat with respect to human health and food security[7,8]. Wet bulb temperature ($T_w$), a synthetical variable combining temperature and humidity, is widely adopted to characterize extreme heat events and limit thresholds are set for survivability (35 °C[9,10],). Heat and humidity together put people at greatly increased risks as elevated $T_w$ hampers body's ability to sweat, potentially leading to heat stroke within a few hours[11–13]. Significant health consequences,

encompassing both morbidity and mortality, materialize at considerably lower $T_w$ values, in contrast to those with conventional air temperature metrics[14–16].

Amplified summertime $T_w$ over past decades was well documented[17,18] and it was projected to further rise in a continuously warming world, particularly in tropical and mid-latitude regions, which are home to roughly half of the world's population[19–21]. As $T_w$ considers the influence of humidity, its variation is fundamentally governed by the interplay between air temperature and atmospheric moisture[18,22]. Notable warming of land surface temperature has been extensively observed since 1900[23], while shifts in atmospheric moisture exhibit considerable spatial heterogeneity, influenced by multiple factors including topography, vegetation, and climate patterns[24–28]. These

[1]Department of Geography, Hong Kong Baptist University, Kowloon Tong 999077 Hong Kong SAR, China. [2]School of Engineering and Applied Sciences, Harvard University, Cambridge, MA 02138, USA. [3]Department of Precision Machinery and Precision Instrumentation, University of Science and Technology of China, Hefei 230026, China. [4]Key Laboratory of Environmental Optics and Technology, Anhui Institute of Optics and Fine Mechanics, Chinese Academy of Sciences, Hefei 230031, China. [5]School of Environmental Science and Optoelectronic Technology, University of Science and Technology of China, Hefei 230026, China. [6]These authors contributed equally: Fan Wang, Meng Gao. ✉e-mail: mmgao2@hkbu.edu.hk; chliu81@ustc.edu.cn

multifaceted influences introduce uncertainties in understanding and attribution of historical and future variations of heat risk[22].

China is one of world's largest countries in terms of land areas where topography, land use and climate exhibit significant inter-regional diversities[29–31]. Owing to its lower latitudes, dense forest cover, and the impact of the East Asia summer monsoon, southern China typically experiences more pronounced heat stress, characterized by higher $T_w$ values, during summer months compared with other parts of the country[32,33]. China's $T_w$ has undergone rapid intensification since 1960s, primarily attributed to human induced climate change[34–36], exerting substantial heat stress on its vast population. However, both rising and falling trends in atmospheric moisture were identified across China. For example, notable decreases in atmospheric moisture were found in South China during 1961–2014[37] and Southwest China during 1979–2013[38], and increasing tendencies were confirmed in the Yangtze River basin during 1961–2005[39]. This indicates that $T_w$ is likely to change differently across regions under climate change, which has not been fully understood. Given the significance of heat stress with respect to human health and food security, a better understanding of how historical and future $T_w$ evolves in different regions and the key driving factors would better assist mitigation and adaptation of heat stress, particularly for populous country like China.

In this study, we report different observed $T_w$ variations over 1979–2018 between northern and southern China, and we find the dominant role of $E_a$ changes in determining the different trend of $T_w$ in southern China from northern China. We associate the heterogeneity of $E_a$ with faster warming of high-latitude regions of East Asia under global warming, which regulates large-scale atmospheric features and leads to extended impacts of the South Asia high (SAH) and the western Pacific subtropical high (WPSH) over southern China and suppressed moisture transport. Attribution analysis using climate model simulations confirm our findings and we further project that the entire eastern China where 94% of China's population live is likely to face widespread and uniform elevated thermal stress under such south-to-north spatially heterogeneous influence at the end of this century. These findings call for development of adaptation measures to avoid adverse impacts of heat stress.

## Results

### Spatiotemporal variations of $T_w$ in China over 1979–2018

The spatial distribution of summertime $T_w$ across China during the period of 1979–2018 reveals a discernible decrease from the south-eastern coastal areas toward inland regions (Fig. 1A). Over the period of 1979–2018, $T_w$ shows increasing trends across the majority of stations, particularly pronounced in northern China where the increasing rate surpasses 0.2 K/decade. However, the rate of increase is notably smaller for southern stations, and decreases in $T_w$ are detected in parts of southern China (Fig. 1B). Surface homogenized humidity products and reanalysis datasets also exhibit similar spatial patterns of $T_w$ trend

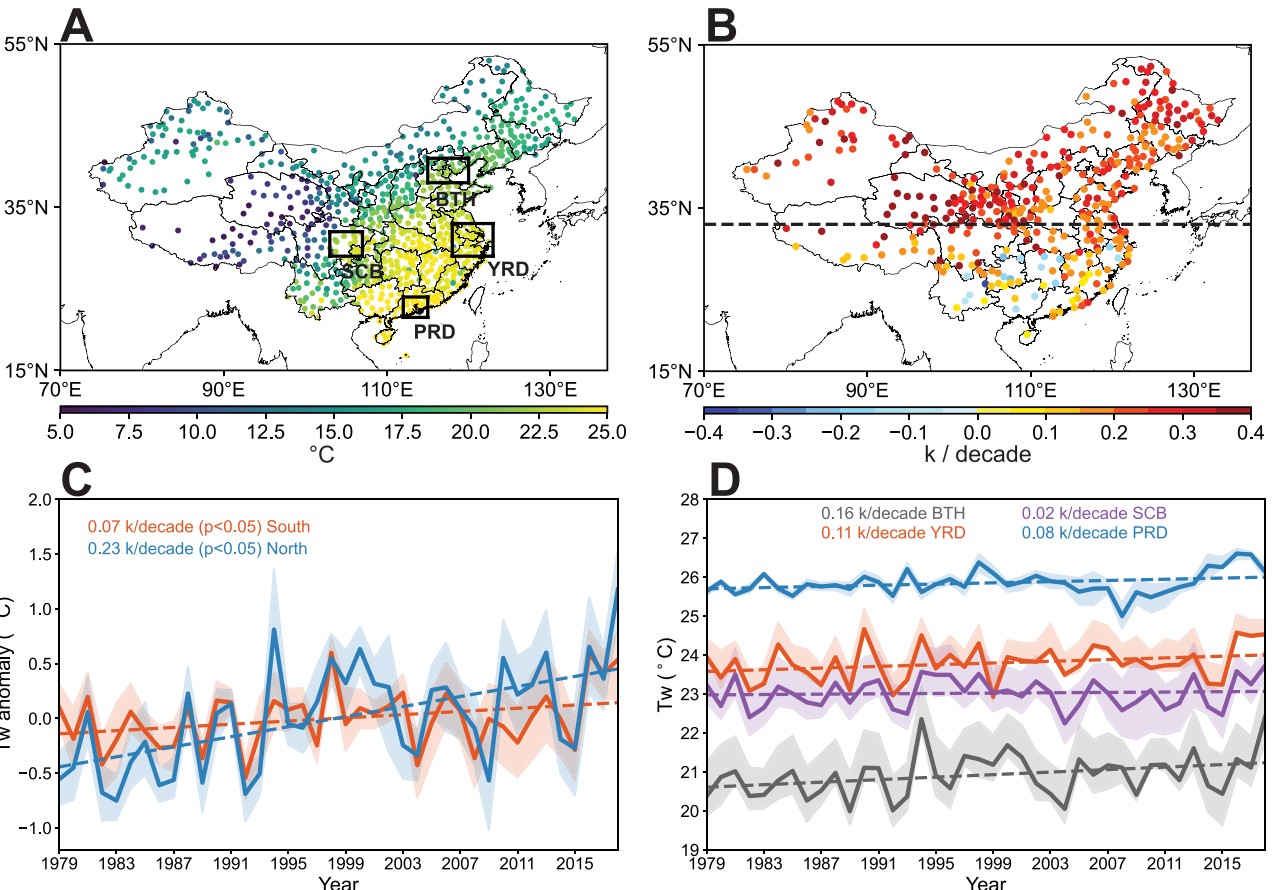

**Fig. 1 | Spatiotemporal variations of wet bulb temperature ($T_w$) in China.**
**A** Spatial distribution of average summertime $T_w$ during the period from 1979 to 2018. Black squares represent four key agglomerations: Beijing-Tianjin-Hebei (BTH, 38°N-41°N, 115°E-120°E), Yangtze River Delta (YRD, 29°N-33°N, 118°E-123°E), Sichuan Basin (SCB, 29°N-32°N, 103°E-107°E) and Pearl River Delta (PRD, 21°N-23°N, 112°E-115°E). **B** Spatial distribution of $T_w$ trend during the period from 1979 to 2018.

Only sites with significant trend ($P < 0.05$) are displayed. Black dashed line indicates the latitude of 33°N. **C** Time series of average $T_w$ anomaly of northern and southern stations during the period from 1979 to 2018. **D** Time series of average $T_w$ of BTH, YRD, SCB and PRD during the period from 1979 to 2018. Source data are provided as a Source Data file.

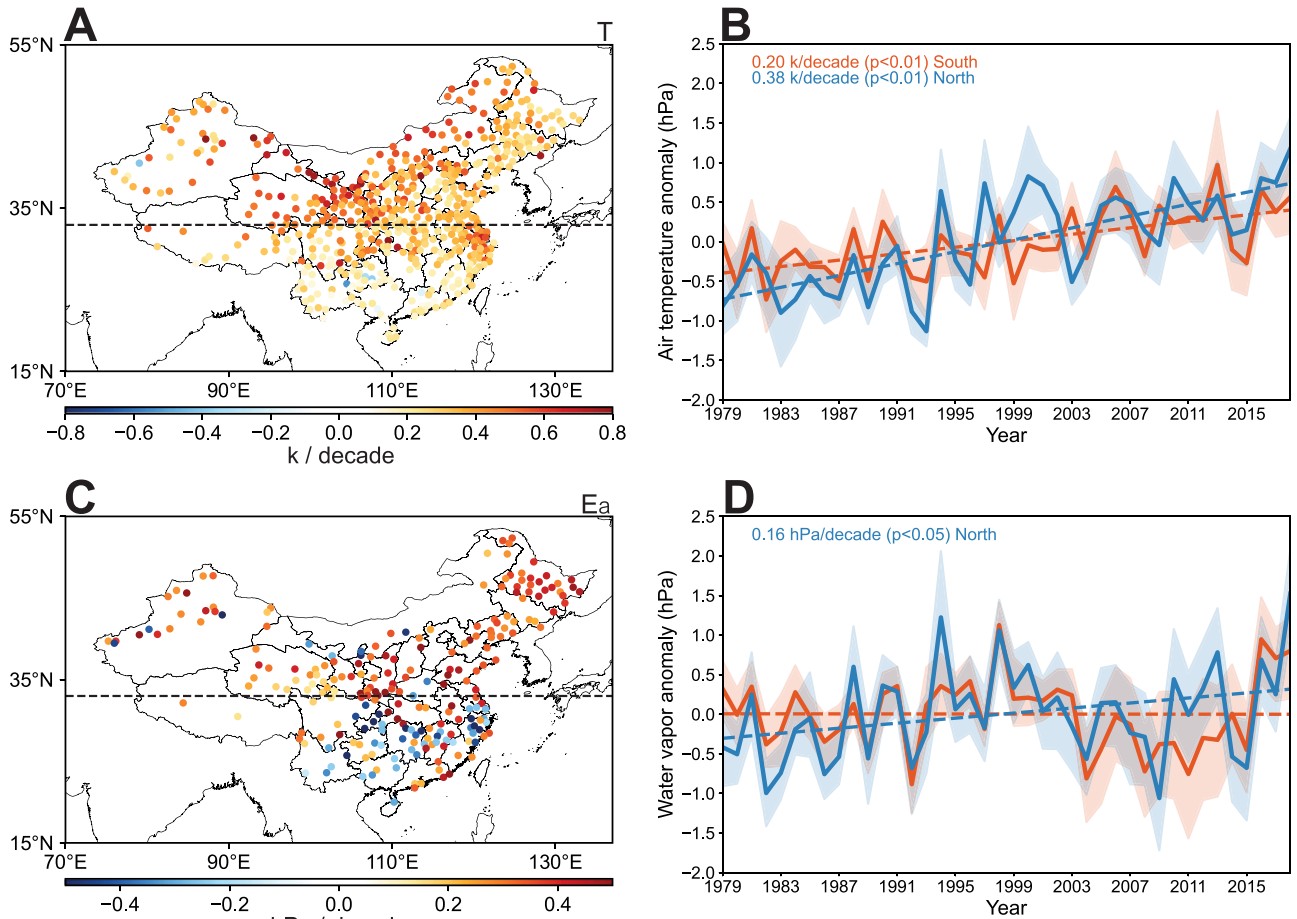

**Fig. 2 | Air temperature (T) and water vapor ($E_a$) variations. A** Spatial distribution of T variations during the period from 1979 to 2018. Only sites with significant trend ($P < 0.05$) are displayed. Black dashed line indicates the latitude of 33°N. **B** Time series of average T anomaly of northern and southern stations during the period from 1979 to 2018. **C** Spatial distribution of $E_a$ variations during the period from 1979 to 2018. Only sites with significant trend ($P < 0.05$) are displayed. Black dashed line indicates 33°N. **D** Time series of average $E_a$ anomaly of northern and southern stations during the period from 1979 to 2018. Source data are provided as a Source Data file.

(see Fig. S1 and S2), confirming the major feature of spatial distributions of $T_w$ variations we find based on surface observations. We categorized stations into southern stations (latitude <33°N) and northern stations (latitude ≥33°N) based on locations relative to the latitude of 33°N since the Qinling-Huaihe Line (32°N-34°N) is commonly used to distinguish between northern and southern China. Regional average $T_w$ for both northern and southern China exhibit statistically significant increasing trends (Fig. 1C). Yet the rate of increasing $T_w$ in northern China (0.23 K/decade) is much higher than that of southern China (0.07 K/decade). Average $T_w$ for four major urban populous agglomerations, namely Beijing-Tianjin-Hebei (BTH), Yangtze River Delta (YRD), Sichuan Basin (SCB) and Pearl River Delta (PRD), also display distinctively upward trends (Fig. 1D). BTH, situated in northern China, exhibits the lowest initial value of $T_w$, whereas experiences the most rapid increase, with a remarkable rate of 0.16 K/decade.

### Key player of faster warming of high-latitudes in changing moisture and attribution analysis

Since $T_w$ is determined by the combined influences of air temperature (T) and actual water vapor pressure ($E_a$), variations in T and $E_a$ can be used to explain $T_w$ variations observed over the preceding decades. Sensitivity of $T_w$ to T and $E_a$ suggests that increasing $E_a$ yields a more pronounced enhancement of $T_w$, particularly evident when T is relatively low (Supplementary Fig. S3). Changes in $T_w$ when either T or $E_a$

varied but the other factor fixed at year 1979 highlight that $E_a$ changes are responsible for the observed different $T_w$ trends in southern China (Fig. 2 and Supplementary Fig. S4). Over the study period of 1979–2018, $E_a$ trends exhibit a distinct North-South divide, characterized by higher enhancement over northern stations (a rate of 0.16 hPa/decade) but a lower rate over southern stations that contracted by declining $E_a$ (Fig. 2C, D). $E_a$ is subject to the influences of several factors including surface evaporation and water vapor dispersion, which in turn can be affected by variables such as surface temperature, land use, precipitation, and atmospheric circulation[40–42]. Over northern China, significant increasing trends in evaporation are identified (Fig. 3A). However, southern China witnesses a decreasing trend in evaporation (Fig. 3A) alongside a concurrent trend of moisture divergence (Fig. 3B). These two factors synergistically lead to a reduction in $E_a$ within that region. Apart from atmospheric processes, land use changes also exert impacts on $E_a$[43]. China's rapid urbanization since 1979 has converted natural land to impervious urban areas, reducing water evaporation[44,45]. The probability density function (PDF) of $E_a$ changes indicates that $E_a$ is more likely to decrease in areas with land use conversion (Supplementary Fig. S5A). Many stations in southern China experienced such conversion during the period of 1979–2018 (Supplementary Fig. S5B), which may partly contribute to $E_a$ decline.

To understand the underlying mechanism and factors causing $T_w$ variations in preceding decades, we conducted an empirical orthogonal function (EOF) analysis on $T_w$ during the period of 1979–2018.

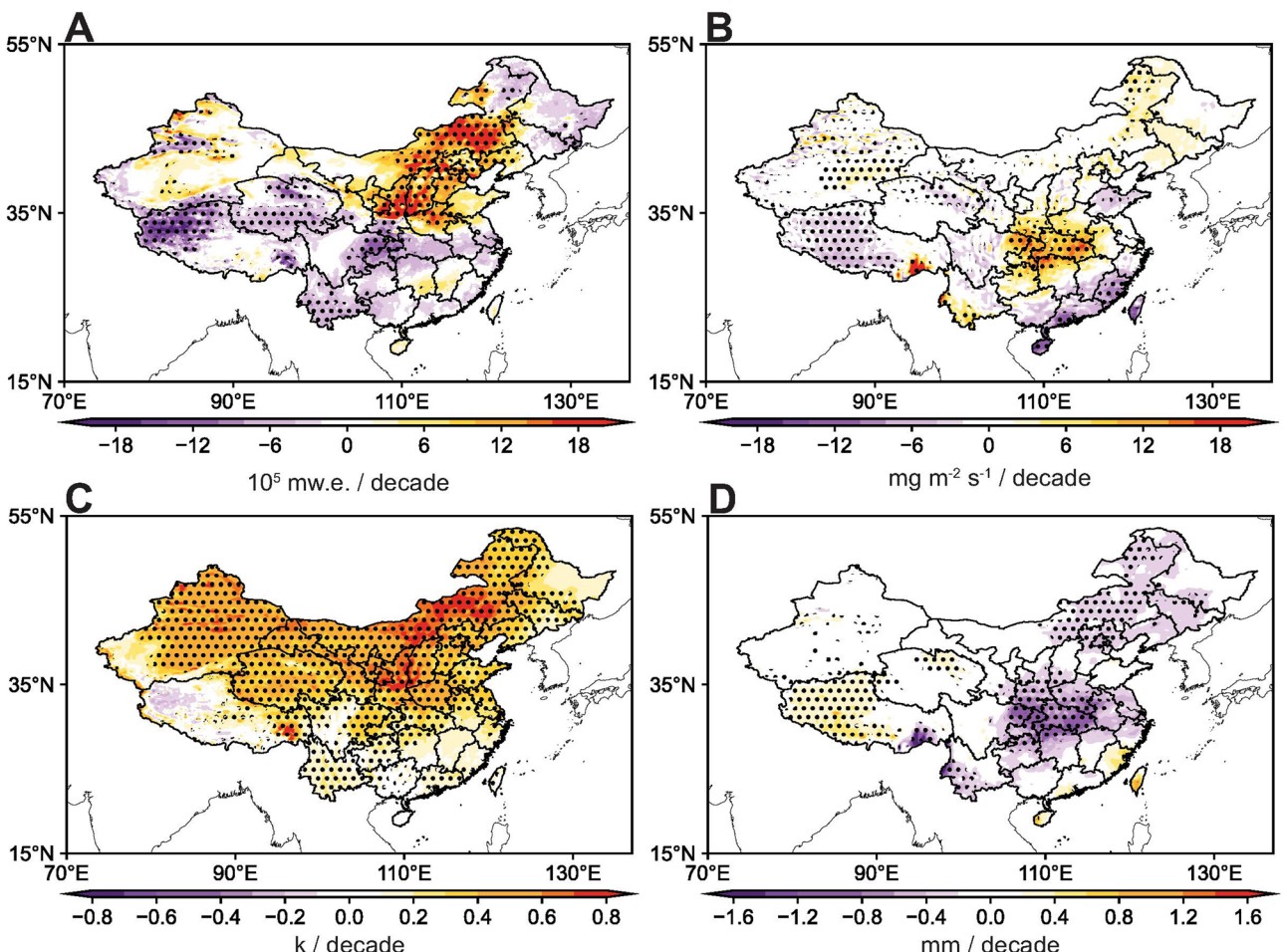

**Fig. 3 | Change in meteorological variables.** Spatial distributions of changes in surface evaporation (**A**), vertical integrated moisture divergence (**B**), surface temperature (**C**) and total precipitation (**D**) over 1979–2018. Black dots denote areas with significant trend ($P < 0.05$). Source data are provided as a Source Data file.

The first leading mode of EOF, accounting for 42.7% of the total variance, shows consistent $T_w$ changes across China with observed trend of $T_w$ variations, characterized by faster increasing rate in northern China (Supplementary Fig. S6A). In line with the significant positive changes revealed by regression of $T_s$ on PC1 (Fig. 4A), $T_s$ also shows significantly increasing trend during the period of 1979–2018 (Fig. 4B) over the high-latitude regions of East Asia. The more rapid warming in this arid region may be due to its low heat capacity[23] and lower anthropogenic aerosol emissions compared to lower latitudes[46]. These observations can be further supported by the findings of surface air temperature changes under GHG-only and aerosols-only forcing outputs from the Coupled Model Intercomparison Project Phase 6 (CMIP6) experiments (Supplementary Fig. S7). It implies that the intensified surface temperature over this region positively contributes to the dominant mode of $T_w$ variations. During the study period, mean surface temperature in summer within the high-latitude regions of East Asia displays a notably increasing trend of 0.51 °C decade$^{-1}$ (Fig. 4C), which is significantly correlated with PC1 (0.73, $p < 0.01$) (Fig. 4D). Faster warming of the high-latitude region results in upper atmospheric pressure heightened, which leads to anticyclone enhancement in northern regions at upper levels (Supplementary Fig. S8A, B). Anticyclone enhancement in northern regions triggers the eastward and northward extension of the South Asia high (SAH, represented by 16760-dagpm line on 100 hPa[47]) (Fig. S9A, B), and westward and northward marching of the western Pacific subtropical high (WPSH, represented by 5880-dagpm line on 500 hPa[48]) (Supplementary Fig. S9C, D), which are typically situated over the Tibetan region and the

Western Pacific Ocean in summer, respectively (Supplementary Fig. S8A, B). Increased influences of the SAH and WPSH over larger swathes of southern China lead to descending motion prevailing over southern China at lower levels, suppressing moisture transport from both the Bay of Bengal and the South China Sea but accelerating moisture transport from the Pacific Ocean to northern China (Supplementary Fig. S8C, D). Correlation and composite analyses further verify the strong connection between $T_w$, and the SAH and WPSH as $T_w$ tends to be higher in northern regions but lower in southern regions when the SAH and WPSH are in higher latitudes (Supplementary Figs. S10 and S11). This results in declined water vapor in the atmosphere, and further suppresses precipitation, leading to less water content of the ground that can be evaporated (Fig. 3D). Historical changes of geopotential height and atmospheric circulation are highly consistent with regression results (Supplementary Figs. S12 and S13), highlighting their contributions to $T_w$ variations.

We further quantified the contributions of different external forcings to $T_w$ changes. Although the CMIP6 historical simulation cannot fully capture certain decreases of $T_w$ in southern China, different increasing rates of $T_w$ between northern and southern China (0.3 K/ decade higher in northern China) can be found (Supplementary Fig. S14B). We find the dominant influence of greenhouse gases (GHG) on variations of $T_w$ (Supplementary Figs. S15 and S16A, approximately 63%), followed by aerosols (AER, ~24%) (Supplementary Figs. S15 and S16B). GHG dominates $T_w$ changes across China, except in the northeastern desert regions (Supplementary Fig. S16E), with a more pronounced effect over South China (Supplementary Fig. S16A). Natural

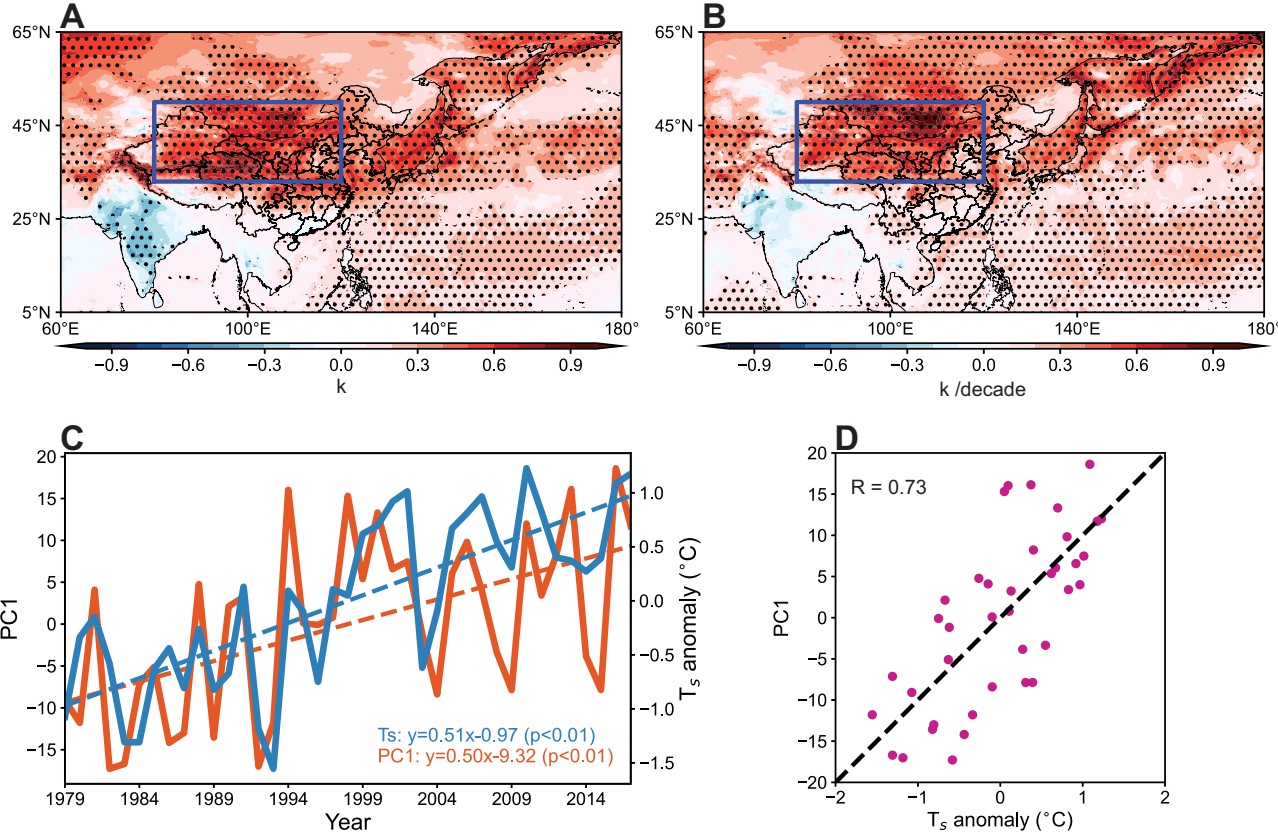

**Fig. 4 | Connection between surface temperature (T_s) and the first leading mode (PC1) of empirical orthogonal function. A** Regression of $T_s$ on the first leading mode. Black dots denote areas with significant correlation ($P < 0.05$). **B** $T_s$ trend over 1979–2018. Black dots denote areas with significant correlation ($P < 0.05$). **C** Time series of PC1 and average $T_s$ anomaly of the northern region in East Asia (NEA, 33°N-50°N, 80°E-120°E, red boxes in A and B). **D** The scatter plot between $T_s$ anomalies in NEA and PC1 in each year from 1979 to 2018. Source data are provided as a Source Data file.

factors (NAT) and land use (LU) play a relatively minor role (Supplementary Fig. S16C, D). In line with Supplementary Fig. S5, LU exerts a negative influence on $T_w$ variations due to urbanization and reduced evaporation, particularly notable in southern regions (Supplementary Fig. S14F). We also employed the empirical decomposition method (EMD) on $T_w$ variations first to separate internal natural variability and external anthropogenic forcing (Supplementary Fig. S17), and then applied the EOF decomposition on external signal. Similar contributions of GHG (-68.6%) and AER (-23.8%) to $T_w$ variations were found (Supplementary Fig. S18).

**Widely-spread and uniformly elevated future heat stress in China**

Given the advantage of high resolution and better regional details, WRF-Chem downscaling model results were used for future projections of heat stress in China. Consistent with historical observations, projected $T_w$ in southern China in Hist period is notably higher than that in other parts of China (Fig. 5A). However, uniform high heat stress is projected for the entire eastern China by the end of the century under both the SSP2-4.5 and SSP5-8.5 scenarios (Fig. 5B, C). These changes are caused by the accelerated increase of $T_w$ in northern regions (Fig. 5D, E). Under the SSP2-4.5 scenario, the most substantial enhancement (by approximately 4–5 °C) of $T_w$ occurs in the North China Plain (NCP), higher than those of ~2–3 °C in southern regions (Fig. 5D). Under the SSP5-8.5 scenario, $T_w$ in the NCP reaches a level comparable to that of traditionally warmer areas like the PRD (Fig. 5C). This phenomenon results from a substantial difference in the increase of $T_w$ between the NCP and the PRD. The rise in $T_w$ within the NCP is approximately 3 °C greater than that of

southern China (Fig. 5E). Spatial distribution of $T_w$ trends across whole China from the CMIP6 experiments adopted in the attribution analysis are consistent with downscaling simulations (Supplementary Fig. S19A−C). Average changes of $T_w$ from all 12 CMIP6 models are $2.29 \pm 0.69$ °C and $2.02 \pm 0.49$ °C under the SSP2-4.5 scenario (Supplementary Fig. S19D, E); $4.64 \pm 1.31$ °C and $4.17 \pm 1.25$ °C under the SSP5-8.5 scenario (Supplementary Fig. S19F, G) in northern and southern China, respectively. Although changes of $T_w$ under the SSP2-4.5 and SSP5-8.5 scenarios from the CMIP6 are relatively smaller compared to WRF-Chem simulations due to its limited representation of spatial details on local scales[49,50], they display similar spatial diversity that $T_w$ in northern China increases faster than that in southern China.

In accordance with historical trends, simulated increases in T display a subtle absence of distinction between northern and southern regions (Supplementary Fig. S20A, B). In contrast, there are notable spatial differences in $E_a$ changes, characterized by a more pronounced elevation in northern China (Supplementary Fig. S20C, D). This phenomenon can be attributed to anticipated shifts in atmospheric systems (Supplementary Figs. S21 and S22). Under a warming climate, both SAH and WPSH exhibit substantial intensification, with a notable increase in central pressure (Supplementary Fig. S21). This leads to zonal nearing of these two systems and weakened tropical easterly jet[51] but accelerated westerly winds over subtropical regions at upper troposphere[52,53] (Supplementary Fig. S23C, D). The accelerated westerly wind enhances the eastward propagation of Kelvin waves[54,55] and convergence of cross-equatorial flows[56,57]. As a result, the descending motion is intensified over tropical regions[58] and consequently anomalous anticyclone is found over the Bay of Bengal and South

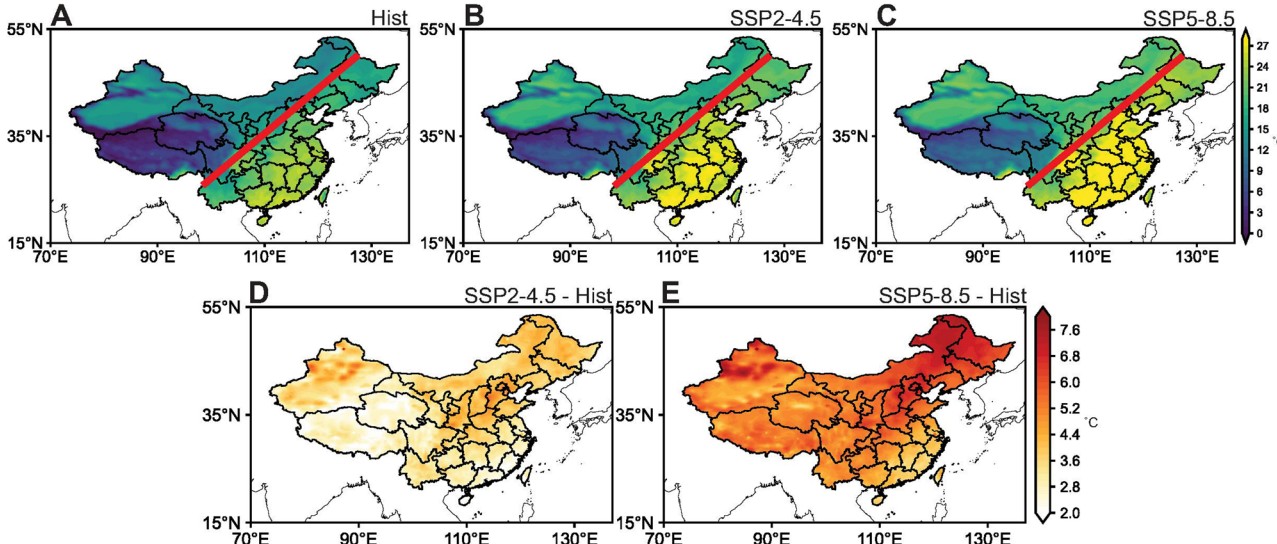

**Fig. 5 | Spatial distribution of wet bulb temperature ($T_w$) and its future shifts.** Spatial distribution of summertime average $T_w$ over 2010-2014 (Hist) (**A**) and over 2096-2100 under the SSP2-4.5 (**B**) and SSP5-8.5 (**C**) scenarios from WRF-Chem simulations. The red line indicates the location of the Heihe−Tengchong Line (and internationally as the Hu line) that divides the area of China into two parts with contrasting population densities. Spatial distribution of differences in summertime average $T_w$ between SSP2-4.5 and Hist (SSP2-4.5 - Hist) (**D**) and SSP5-8.5 and Hist (SSP5-8.5 - Hist) (**E**) from WRF-Chem simulations. Source data are provided as a Source Data file.

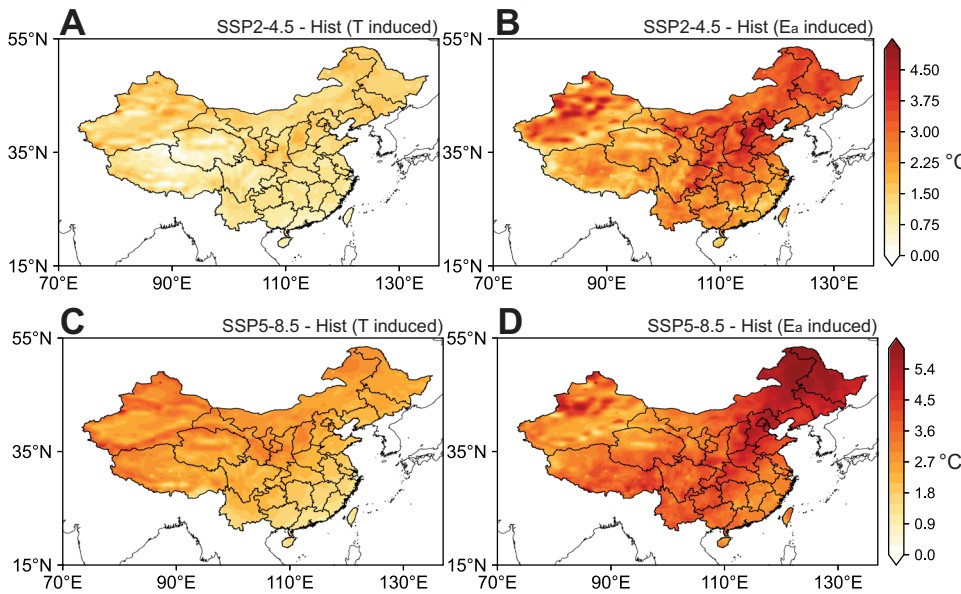

**Fig. 6 | Spatial distribution of air temperature (T) induced and water vapor ($E_a$) induced changes of wet bulb temperature ($T_w$).** Spatial distribution of T induced (**A**) and $E_a$ induced (**B**) changes of $T_w$ under the SSP2-4.5 scenario (SSP2-4.5 - Hist) from WRF-Chem simulations. Spatial distribution of T induced (**C**) and $E_a$ induced (**D**) changes of $T_w$ under the SSP5-8.5 scenario (SSP5-8.5 - Hist) from WRF-Chem simulations. Source data are provided as a Source Data file.

China Sea at lower troposphere (Supplementary Fig. S23E−H). Similar atmospheric circulation anomaly has been observed during the extreme dry-wet contrast event between southern and northern China in 2020[59,60]. Accordingly, influenced by the air flow at the ridge of anticyclone, more water vapor is transported to the Pacific Ocean instead of to southern China, different from the enhanced moisture transport from the Pacific Ocean to northern China (Supplementary Fig. S23). Under both the SSP2-4.5 and SSP5-8.5 scenarios, the increase in $T_w$ resulting from changes in $E_a$ is notably more substantial than that by T across most regions of China (Fig. 6). The impact of elevated $E_a$ contributes roughly twice of the influence of T to increased $T_w$, particularly in humid and semi-humid areas.

## Discussion

$T_w$ has gained considerable attention as a pivotal metric to characterize heat risks on public health[61–63]. Yet spatial heterogeneity of historical and future $T_w$ variations in China and the underlying reasons have not been well understood. In this study, we revealed the spatially differentiated historical $T_w$ changes in China, characterized by a faster rise of $T_w$ and a more pronounced aggravation of heat stress in northern China. We found the dominant role of changes in $E_a$ and attributed such heterogeneity to declining $E_a$ across most of the southern areas, which was associated with faster warming in high-latitude regions of East Aisa. A faster warming of high-latitudes of East Asia regulates atmospheric features, leading to extended impacts of

the SAH and WPSH over southern China and suppressed moisture transport under global warming. Dynamical downscaled future projections further demonstrate the findings from historical records that future $T_w$ variations will mainly be regulated by changes in $E_a$, which exhibit south-to-north spatial heterogeneity under a warming climate. Accordingly, the entire eastern China where 94% of China's population live is likely to experience widespread and uniform elevated thermal stress at the end of this century, particularly under the SSP5-8.5 scenario. These findings necessitate development of adaptation measures to avoid adverse impacts of heat stress, in addition to investments in renewables.

While multi-model ensembles (MME) data helps mitigate internal climate variability, the influence of low-frequency variability[64], such as the Interdecadal Pacific Oscillation (IPO) and the Atlantic Multidecadal Oscillation (AMO), could still impact the long-term trend of $T_w$ in attribution analysis due to the relatively short period considered in this study[65]. The alignment of EMD analysis (Supplementary Fig. S17) and EOF decomposition (Supplementary Fig. S18) on observed $T_w$ variations with the results obtained from attribution analysis (Supplementary Fig. S15) underscores the relatively small impact of internal climate variability on $T_w$ variations during the study period, indicating minor impacts on main findings. Our study is still limited by computational resources. A five-year average of downscaled future climate could be influenced by internal climate variability, leading to uncertain projections. Using data from the CMIP6 experiments listed in Supplementary Table S1, we reveal significant increasing trends in $T_w$ across China under the SSP2-4.5 and SSP5-8.5 scenarios from 2015 to 2100 (Supplementary Fig. S24). Notably, we observe a pronounced south-north difference, particularly under the SSP5-8.5 scenario, indicating that the non-uniform enhancement of $T_w$ due to global warming is reasonably robust. Regional downscaling covering longer decades would contribute to better projections for China if computational resources permit.

Our study holds several important implications for understanding the evolving dynamics of $T_w$ and its driving factors in China. The acceleration of $T_w$ increases in northern China challenges the conventional understanding of China's regional summer climates, which historically categorized the south as hot and humid and the north as dry[66–68]. This shift in perception holds significant implications for climate adaptation and policy development, prompting a reevaluation of strategies to address evolving heat stress patterns. The examination of East Asia's climatic systems' impact on regional climate patterns provides valuable insights into the larger-scale mechanisms that shape $T_w$ trends. Our findings contribute to the broader understanding of climate change's intricate impacts on heat stress in a specific geographic context.

By dissecting the interplay between T and $E_a$, our study sheds light on the important influence of $E_a$, particularly under specific temperature conditions. This understanding is crucial for accurate heat stress assessments and efforts of disaster preparedness. Under the background of global warming, China confronts an ongoing rise in heat extremes[69–72]. The elevation of air temperature is accompanied by a surge in the occurrence of compound extremes, including extreme heat wave coupled with drought[73–75], regional compound heat and precipitation extremes[76–78], and cooccurrence of heat wave and ozone pollution[62,79]. A profound understanding of dominant factors in these heat-induced natural disasters is pivotal in devising more effective strategies to mitigate and address these challenges.

## Methods
### Observations, reanalysis and land use data
Surface observations of daily air temperature (T) and relative humidity (RH) at 2 meters from 1979 to 2018 were obtained from the weather stations maintained by the China Meteorological Administration

(CMA). We selected this study period because the number of surface weather stations of used dataset increased notably after 1979 (Supplementary Fig. S25A), and there was a shift in the trend of $T_w$ variations from a decreasing trend to an increasing trend after the late-1970s (Supplementary Fig. S25B), largely due to global warming indicated by surface temperature anomaly since late-1970s (Supplementary Fig. S25C). We calculated actual water vapor pressure based on Eqs. (1) and (2):

$$RH = \frac{E_a}{E_s} \times 100\% \tag{1}$$

$$E_s = 6.112 \times e^{\frac{17.62 \times T}{T + 243.12}} \tag{2}$$

where $E_s$ represents saturated water vapor pressure. Following Stull[10] and Xiao et al.[62], $T_w$ (°C) was calculated using Eq. (3):

$$
\begin{aligned}
T_w = T \cdot atan\left[0.151977(RH + 8.313659)^{\frac{1}{2}}\right] \\
+ atan(T + RH) - atan(RH - 1.676331) \\
+ 0.00391838(RH)^{\frac{3}{2}} \cdot atan(0.023101 \cdot RH) - 4.686035
\end{aligned}
\tag{3}
$$

where T denotes temperature in °C, and RH stands for relative humidity in % format. This formulation is applicable for RH ranging from 5% to 99% and temperature ranging from −20 to 50 °C[10], which are reasonable ranges for this study. $T_w$ calculated with other methods[80,81] showed comparable spatial distributions of average and variation over the period of 1979–2018 (Supplementary Fig. S26). Using Eqs. (1)–(3), we obtained T and $E_a$ sensitivity by calculating $T_w$ with T and $E_a$ ranging from 10 °C to 40 °C and 10 hPa to 40 hPa, respectively. When $E_a$ exceeded the saturated water vapor pressure, RH was set to 100%.

Meteorological variables including monthly gridded surface air temperature, surface dew point, evaporation, total precipitation, geopotential height and u- and v-components of wind at 100 hPa and 500 hPa, vertical integrated moisture divergence for the same time period were obtained from the European Centre for Medium-Range Weather Forecasts (ECMWF) Reanalysis version 5 (ERA5) dataset[82]. The spatial resolution of these variables was 0.25° × 0.25°.

Homogenized daily surface relative humidity from Li et al.[83] and surface air temperature from Argiriou et al.[84] were employed to verify the trend of $T_w$ and $E_a$ derived from the original observations. We also adopted monthly $T_w$ and $E_a$ from the Met Office Hadley Centre Integrated Surface Database Humidity (HadISDH, https://www.metoffice.gov.uk/hadobs/hadisdh). HadISDH is a homogenized and quality controlled 5° × 5° gridded global surface humidity dataset[85]. It is derived from a variety of sources, including weather station observations, satellite measurements, and reanalysis models. Daily maximum $T_w$ at 1834 stations that had passed quality control and been homogenized were obtained from the global station-based daily maximum wet-bulb temperature (GSDM-WBT) data provided by Dong et al.[86]. Specific humidity and air temperature at 2 meters from the Modern-Era Retrospective analysis for Research and Applications, Version 2 (MERRA2)[87] were adopted to calculate $E_a$ and $T_w$. The spatial resolution of these variables was 0.625° × 0.5°.

Land use dataset in China was obtained from National Tibetan Plateau Data Center (https://data.tpdc.ac.cn). The data generation relied on Landsat Thematic Mapper ™ /Enhanced Thematic Mapper (ETM) Remote Sensing Images, and it was produced through manual visual interpretation. The data set included seven periods: the end of 1980s, 1990, 1995, 2000, 2005, 2010 and 2015. The spatial resolution of this dataset was 30 m. We determined conversion time when the first time the station location grid changed to urban land use.

## Statistical analysis

We adopted EOF analysis to decompose spatiotemporal variations of $T_w$ during the period of 1979–2018 in China. Student T-test was used to detect the statistical significance of the regression coefficient and correlation coefficient. We also employed EMD decomposition on $T_w$ variations to separate internal natural variability and external anthropogenic forcing. EMD is a data-adaptive multiresolution technique to decompose non-linear and non-stationary signals by separating them into physically meaningful components at different resolutions[88]. Decomposed signals can be considered as periodic oscillations with different frequencies[89,90], while the residual can capture impacts of external forcings[91]. Additionally, we conducted EOF analysis on external signals to obtain the dominant patterns or modes of variability and linked them to the contributions of each anthropogenic factor including GHG, AER and LU.

We also conducted composite analysis on latitude and longitude of the SAH and WPSH of years when extreme northward/southward and westward/eastward movements of the SAH and WPSH occurred. The extreme movements of the SAH and WPSH were defined as those latitudes and longitudes that were larger than one standard deviation. The PDFs for changes of $E_a$ in areas with and without land use conversion were computed based on normal distribution using the following equation,

$$f(x) = \frac{1}{\sigma\sqrt{2\pi}} e^{-\frac{(x-\mu)^2}{2\sigma^2}} \qquad (4)$$

where $\sigma$ is the sample standard deviation and $\mu$ is the mean value.

## CMIP6 historical simulations and attribution analysis

The historical simulations of monthly T and RH driven by time evolving all forcings and single forcing from the Detection and Attribution Model Intercomparison Project (DAMIP)[92] and the Land Use Model Intercomparison Project (LUMIP)[93] in the Coupled Model Intercomparison Project Phase 6 (CMIP6) were employed in this study to assess climate response to individual forcings, including GHG-only (GHG), aerosols-only (AER), natural-only (NAT) and land use-only (LU). We used a total least-squares optimal fingerprinting approach to attribute the contribution of each single forcing to variations of $T_w$ under all forcing conditions[94,95]. It is a generalized linear regression model to portray variations of $T_w$ as a linear amalgamation of alterations induced by GHG, AER, NAT and LU. The regression model is:

$$T_{all} = \beta_{GHG}T_{GHG} + \beta_{AER}T_{AER} + \beta_{NAT}T_{NAT} + \beta_{LU}T_{LU} + \varepsilon \qquad (5)$$

where $T_{all}$ is a vector of variations of $T_w$ caused by all forcings, and $T_{GHG}$, $T_{AER}$, $T_{NAT}$ and $T_{LU}$ are estimates of the responses to GHG, AER, NAT and LU forcing, respectively. The $\beta$ terms stand for the corresponding scaling factors, and $\varepsilon$ is the residual variability generated internally in the climate system. As most models only provide outputs ending in 2014, the historical simulations over the period of 1979–2014 were considered. We calculated the ensemble mean of runs contributed by each single model and then calculated the average of all models to ensure equal weighting for each model in this analysis. The model outputs were interpolated to the resolution of 1° × 2° using liner interpolation, and the list of selected models is shown in Supplementary Table S1.

## Dynamical downscaling of current and future $T_w$ in China

In this study, we also used the Weather Research and Forecasting model coupled with Chemistry (WRF-Chem) version 3.6.1 to predict current and future regional climate of China with horizontal grid resolutions of 36 km × 36 km. The chemistry of air pollutants and their interactions with regional climate were also included. Gas-phase and aerosol chemistry were modeled by the Carbon-Bond Mechanism version Z (CBMZ)[96] and the 8-bin version of the Model for Simulating Aerosol Interactions and Chemistry (MOSAIC)[97]. To address underprediction of sulfate in China, heterogeneous reactions were integrated based on Gao, et al.[98]. Other physical parameterizations were aligned with those in Gao et al.[99], including Lin microphysics[100], RRTM long wave radiation[101], Goddard short wave radiation[102], Noah land surface model[103], and the Yonsei University planetary boundary layer parameterization[104].

The historical simulation spanned five years from 2010 to 2014 (referred to as Hist). Meteorological initial and boundary conditions for Hist were derived from the 6-hourly National Centers for Environmental Prediction (NCEP) FNL (Final) analysis data (https://rda.ucar.edu/datasets/ds083-2/), provided at a resolution of 1.0° × 1.0°. For the future simulation, a timeframe of five years, ranging from 2096 to 2100, was considered for both the SSP2-4.5 and SSP5-8.5 scenarios[105]. SSP2-4.5 represents a medium pathway for future greenhouse gas emissions and incorporates an additional radiative forcing of 4.5 W m$^{-2}$ by 2100. SSP5-8.5 symbolizes a fossil-fueled development trajectory, accompanied by intensified fossil fuel exploitation and an energy-intensive global lifestyle, with an additional radiative forcing of 8.5 W m$^{-2}$ by the year 2100. Meteorological initial and boundary conditions for the future downscaling simulation were sourced from a bias corrected CMIP6 global dataset, encompassing 18 models from the CMIP6 under the SSP2-4.5 and SSP5-8.5 scenarios[106]. Chemical initial and boundary conditions were maintained at 2010 levels using Community Atmosphere Model with Chemistry (CAM-chem) model outputs[107]. To enhance the accuracy of simulated meteorological variables, the four-dimensional data assimilation (FDDA) technique was adopted, effectively nudging horizontal winds, temperature, and moisture across all vertical levels. We used grid nudging and the adopted nudging coefficients for these variables were all 0.0003.

Current and future anthropogenic air pollutant emissions and biomass burning emissions were taken from O'Neill et al.[105]. To ensure coherence with the chosen future scenarios, we used anthropogenic air pollutant emissions corresponding to the SSP2-4.5 and SSP5-8.5 scenarios for respective simulations. Biogenic emissions were computed in real-time using the Model of Emissions of Gases and Aerosols from Nature (MEGAN)[108]. Current and future $T_w$ variations in China were then computed using Eq. (3) based on WRF-Chem model outputs.

## Data availability

All data are available in the main text or the supplementary information. Source data generated in this study are provided in the Source Data file. Source data are provided with this paper.

## Code availability

The WRF-Chem model is open access at https://github.com/wrf-model. Code to reproduce the findings is available at https://github.com/luka418/moisture-paper.

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

## Acknowledgements

This study was supported by the grants from National Natural Science Foundation of China (no. 42322902), National Key Research and Development Program of China (2022YFC3700103), and the Research Grants Council of the Hong Kong Special Administrative Region, China (project nos. C2002-22Y, HKBU22201820 and HKBU12202021) (MG).

## Author contributions

M.G. and M.B.M. designed the study, and data were prepared by C.L. and R.Z.; F.W. conducted data analyses with contributions from R.Z. F.W. and M.G. wrote the paper with inputs from all authors.

## Competing interests

The authors declare no competing interests.
