## [Peer Review File · Nature Communications]

Uniformly elevated future heat stress in China driven by spatially heterogeneous water vapor changesREVIEWER COMMENTS

Reviewer #1 (Remarks to the Author):

Review of “Widely-spread and uniformly elevated future heat stress in China driven by spatially heterogeneous response of water vapor” by Wang et al.

This study has explored the underlying mechanism of the south-to-north spatially heterogeneous changes of the wet bulb temperature (Tw) in China as well as its associated future change using the several datasets, including site-observations, reanalysis and CMIP6 model simulations. Results revealed that the extension of the South Asia high and the western Pacific subtropical high are the main drivers for Tw change in southern China. It further indicated that China is likely to experience the widespread and uniform elevated thermal stress across China in response to future warming. This topic is hot currently and is valuable for exploration. However, the analyses in this study included too many points, involving the change in Tw, sensitivity of Tw to T and Ea, underlying mechanisms, attribution analysis, and future projection. There are too much text and there is no focusing point. Additionally, the analyses and findings in this study seem to be quite similar with the previous studies. So, I cannot recommend it to be accepted. More specific comments can be found in the following.

1. Line 61: Is China the second-largest population now?
2. Line 83: “increasing” may be better than “enhancement”
3. Line 90: The regions’ names are needed to be labeled in Fig. 1A
4. Line 97-: Here, the authors have investigated the sensitivity of Tw to T and Ea. They indicated that the changes of Tw in northern China and southern China are mainly caused by changes in Ea. How do you get this conclusion? From Fig. 2, I think the role of temperature seems to be much greater than Ea. If the conclusion you obtained is ok, more evidence are needed, for example, how many contribution from T or Ea should be clear.
5. Line 130-: How to identify the time of the land use conversion for each station.
6. In this study, the authors have evaluated contribution of GHG, AER, NAT, and LU to the change in Tw. It is ok. I want to know whether these signals can be detected for the observed change in Tw in China. But I have not found any information from the current MS.
7. Figure S8: The GHG presents a uniformly influence on Tw across China. But there is almost reversing impact of AER over northern and southern China. Why?
8. Line 143: Note the difference between “predict” and “project”
9. “SSP245” and “SSP585” should be “SSP2-4.5” and “SSP5-8.5”
10. Line 151: Why the rise in Tw in northern China shows much greater than that in southern China in the future?
11. Line 155: Is “northern China” ok?
12. Figure S10: The change patterns are much better than the climatic means.
13. Line 167: This description is not correct and there have been many such works.
14. Line 209: There are many calculations for Tw. Their difference may be need for discussion
15. Line 212: “due point”?
16. Line 243: In general, more runs are much better for such analysis

17. Line 244: Which method is used for the interpolation? Is the topographical effect considered for the temperature interpolation?

18. Line 246-: I wonder why the authors used downscaling simulations for the future projection. Additionally, the models used for the drivers of the downscaling are different from the early section of attribution analyses, which seem to be much arbitrary for the analysis in this study and the conclusions may be inconsistent across this MS.

Reviewer #2 (Remarks to the Author):

Comments on the manuscript entitled "Widely-spread and uniformly elevated future heat stress in China driven by spatially heterogeneous response of water vapor" by Wang et al submitted to Nature Communications.

The authors investigated the changes in wet bulb temperature (T_w) using observed temperature and relative humidity data over the period 1979-2018. They have found that T_w has experienced a more significant increase in northern China compared to southern China, primarily due to spatially heterogeneous changes in water vapor pressure. Notably, such changes can lead to heightened thermal stress, thereby potentially impacting human health and adaptation measures in eastern China. This study is likely to generate immediate interest among professionals in climate- and health-related disciplines.

Nevertheless, there are some limitations to address. This study only used the observational data after 1979. The observational data can be extended back to 1960 in China. The absence of homogenization of the raw observational data could influence the findings. Furthermore, the dynamical downscaling simulations and mechanism analysis presented in the manuscript do not offer sufficient support for the stated conclusions.

Given these limitations, I recommend substantial revisions before considering this manuscript for publication in Nature Communications. A homogenized observational data, along with additional evidence from climate model simulations and mechanism analysis, is necessary to strengthen the study's validity and impact:

Major comments:

1. Data: The clarity of data sources and processing methods is crucial for the robustness of the study. The authors should address the following points:
 - (a) Clarify whether the observational data and CMIP6 data used are "daily" or "monthly."
 - (b) Please justify utilizing only observational data after 1979. Note that daily temperature and relative humidity data in China since 1960 have been accessible.
 - (c) Temperature and relative humidity data should undergo homogenization, as many observational data

contain breakpoints resulting from automated observation implementation and/or station relocation, etc. These may substantially affect the trend of temperature and relative humidity and lead to unreliable conclusions. The authors may consider using homogenized daily temperature and relative humidity data in China since 1960 to enhance data reliability (Argiriou, et al, 2023; Li et al, 2020).

2. Method: The methodology section requires additional clarification/additional numerical simulation and estimation of uncertainty in Tw projection. Address the following concerns:

(a) The authors conducted a set of 36km WRF-Chem simulations over the historical (2010-2014) and future (2096-2100) periods to investigate the future changes in Tw. However, the historical simulation was driven by FNL analysis data, while the future simulation was driven by bias-corrected CMIP6 data. Thus, the difference in WRF-Chem simulations between future and historical periods can be partly attributed to different sources of large-scale forcing data rather than climate change alone.

(b) Specify if the projections in Figs. 5, 6, S9, and S10 are based on the WRF-Chem simulations or the raw CMIP6 datasets.

(c) Examine whether CMIP6 and WRF-Chem simulations exhibit similar trends in Tw. The spread among individual CMIP6 simulations and WRF-Chem is essential for estimating the uncertainty of changes in Tw, Ea, and T. It is necessary to include an analysis of the uncertainty associated with the Tw projection using different sources of data.

3. Mechanism: The authors must provide a more comprehensive understanding of the mechanisms driving historical and future changes in water vapor pressure (Ea) and wet bulb temperature (Tw).

Consider addressing the following concerns:

(a) The authors attributed the historical change in water vapor pressure (Ea) and wet bulb temperature (Tw) to the changes in the South Asian High (SAH) and Western Pacific Subtropical High (WPSH). However, in addition to the SAH and WPSH, other factors, e.g. the Asian summer monsoon, ENSO, and PDO, also play important roles in modulating the East Asian climate. Please elaborate on the extent to which the SAH and WPSH influence Ea and Tw compared to other factors.

(b) The provided explanation regarding the influence of the SAH and WPSH on Ea and Tw lacks persuasiveness. It is imperative to conduct additional investigations to elucidate the role of external forcing and internal climate variability in modulating changes in Tw. Please refer to the detailed comments provided in the following section for further insights.

Other comments:

L55: replace "country" with "countries"

L86-89: The criteria for categorizing stations into northern and southern stations using 33N are unclear. Considering the observed tri-pole pattern of summertime precipitation over eastern China in the second half of the 20th century, particularly in the North China-Yangtze River Basin-South China regions (e.g. Ding et al., 2008), further details on the rationale for station classification are important to enhance the robustness of the analysis. To my knowledge, the precipitation trend shows a dry (North China)-wet (Yangtze River Basin)-dry (South China) pattern over eastern China in summer over the past 60 years or so. Why does the water vapor pressure (Ea) increase in North China and decrease in South China (Fig. 2c)?

L99-100: The authors concluded that E_a plays a more important role than T in determining the increase in T_w . However, the means of establishing this relative importance remain unclear. Comparison of Fig. 1B, Fig. 2A, and Fig. 2C suggests that T , conversely, appears to play a more pivotal role. Both T and T_w show a widespread increase in China with a pronounced trend in northern China and a less pronounced trend in the south.

L112-113: The summertime climate in East Asia is influenced not only by the South Asia High (SAH) and the Western Pacific subtropical High (WPSH) but also by factors such as the East Asian summer monsoon, ENSO, and PDO.

L118-127: I'm not convinced by the author's rationale for the decline in water vapor over southern China linked to changes in SAH and WPSH. Notably, the most substantial increases in geopotential height occur beyond the core regions of SAH and WPSH, around 45N and 60N. The eastward movement of SAH and the westward shift of WPSH, as depicted in Fig. S3, are not clearly discernible. Global warming induces an overall elevation in geopotential height; hence, a marginal increase in East Asia cannot be equated with an eastward or westward shift of high-pressure systems. Furthermore, atmospheric circulation is closely related to the gradient of geopotential height (GH), not GH itself. The overall GH increase may not affect the GH gradient, circulation, or water vapor divergence. Additionally, given that most atmospheric moisture resides below 700hPa, the focus on mid and upper-tropospheric changes rather than the lower troposphere lacks justification. Lastly, it is crucial to explore if the eastward ridge point of SAH and the westward ridge point of WPSH in Fig. 4 significantly correlate with T_w changes.

L143: replace "predicted T_w " with "projected T_w ". Please note the difference between "prediction" and "projection".

L155: The expression "divergence in E_a " is confusing, a rephrasing is necessary for clarity.

L157-158: Again, an increase in pressure does not inherently influence circulation; it is the pressure gradient that holds significance. The authors attempt to elucidate the change in E_a (2-m water vapor pressure) using Figs. S10 and S11, where 100hPa and 500hPa pressure/circulation are showcased. It is advisable to investigate lower tropospheric circulation and moisture transfer, as an overall increase in GH might not impact the GH gradient, circulation, or water vapor divergence.

Figs. 1-4: Clarify the significance test employed. Is the serial correlation considered in the significance test?

Fig. 5D, 5E, Fig.S9: These figures illustrated the difference in T , E_a , T_w between 2096-2100 and 2010-2014. A five-year average might be strongly influenced by internal climate variability, amplifying projection uncertainty. Recommending the use of a 30-year average or a long-term trend for more robust projections.

Fig. S2: Clarify the method of obtaining T and E_a sensitivity. Eliminate unrealistic configurations where E_a

exceeds saturated water vapor pressure (E_s) from the figure to avoid misleading.

Fig. S5: Provide technical details on how the Probability Density Functions (PDFs) were derived. How did the authors exclude the impacts other than land use change? Explain how the urban and rural stations are identified in the text.

Figs. S6-8: Do the authors refer to “anthropogenic” aerosol experiment?

References and dataset

Argiriou A A, Li Z, Armaos V, Mamara A, Shi Y L, Yan Z W. 2023: Homogenised Monthly and Daily Temperature and Precipitation Time Series in China and Greece since 1960. *Adv. Atmos. Sci.*, doi: 10.1007/s00376-022-2246-4

Li, Z., Z. W. Yan, Y. N. Zhu, N. Freychet, and S. Tett, 2020: Homogenized daily relative humidity series in China during 1960–2017. *Adv. Atmos. Sci.*, 37(4), 318–327, <https://doi.org/10.1007/s00376-020-9180-0>

Daily temperature and relative humidity data link

<https://doi.org/10.57760/sciencedb.01731>

<https://doi.org/10.11922/sciencedb.804>

Ding et al. (2008) Inter-decadal variation of the summer precipitation in East China and its association with decreasing Asian summer monsoon. Part I: Observed evidences. *Int. J. Climatol.* 28, 1139–1161.

Ding et al. (2009) Interdecadal variation of the summer precipitation in China and its association with decreasing Asian summer monsoon. Part II: Possible causes. *Int. J. Climatol.*, 29, 1926–1944, doi:10.1002/joc.1759

Huang et al. (2023) Relative contributions of internal variability and external forcing to the inter-decadal transition of climate patterns in East Asia. *npj Clim Atmos Sci* 6, 21. <https://doi.org/10.1038/s41612-023-00351-0>

Zhang, Y. et al. (2021) Projections of tropical heat stress constrained by atmospheric dynamics. *Nat. Geosci.* 14, 133–137. <https://doi.org/10.1038/s41561-021-00695-3>

Widely-spread and uniformly elevated future heat stress in China driven by spatially heterogeneous response of water vapor

Tracking #: NCOMMS-23-52750

Authors: Wang et al.

REVIEWER COMMENTS

Response to Reviewers

Reviewer #1:

Comments:

Review of “Widely-spread and uniformly elevated future heat stress in China driven by spatially heterogeneous response of water vapor” by Wang et al.

This study has explored the underlying mechanism of the south-to-north spatially heterogeneous changes of the wet bulb temperature (T_w) in China as well as its associated future change using the several datasets, including site-observations, reanalysis and CMIP6 model simulations. Results revealed that the extension of the South Asia high and the western Pacific subtropical high are the main drivers for T_w change in southern China. It further indicated that China is likely to experience the widespread and uniform elevated thermal stress across China in response to future warming. This topic is hot currently and is valuable for exploration. However, the analyses in this study included too many points, involving the change in T_w , sensitivity of T_w to T and E_a , underlying mechanisms, attribution analysis, and future projection. There are too much text and there is no focusing point. Additionally, the analyses and findings in this study seem to be quite similar with the previous studies. So, I cannot recommend it to be accepted.

Reply:

- Thank you for your time to offer valuable suggestions. We agree that the analyses included changes of T_w , sensitivity of T_w to T and E_a , underlying mechanisms, attribution analysis and future projection. These different parts are actually closely related and are in line with our major objective of this study.
- The major objective of this study is to understand how T_w evolved in the past and will change in the future, given its impact on human health. A better understanding of its historical change will help constrain uncertainties in future projection. Our analyses were organized with a focus on this objective.
- We first would like to understand what drove the historical observed variations of T_w . As T_w is influenced by both T and E_a , we explored whether the driver for the change was T or E_a . We then used climatology and attribution analysis to understand why E_a was changing, and further used future projection to confirm the relationship found in historical records.

- We highlighted the significant influence of water vapor on south-to-north spatially heterogeneous trends of historical and future T_w variations in China. We elucidated that external forcing factors contributed to this phenomenon. The findings from both historical and future analyses support each other, underscoring that those changes in circulation patterns, within the context of a warming climate, are responsible for the widely-spread and uniformly elevated heat stress in China.
- To address your concern, we re-organized the manuscript and added transition sentences to help readers to understand the logic of this study better.
- “The spatial distribution of summertime T_w across China during the period of 1979-2018 reveals a discernible decrease from the southeastern coastal areas toward inland regions (Fig. 1A).”
- “Since T_w is determined by the combined influences of T and actual water vapor pressure (E_a), variations in T and E_a can be used to explain T_w variations observed over the preceding decades.”
- “We further quantify the contributions of different external forcings to T_w changes.”
- “Consistent with historical observations, projected T_w in southern China in Hist period is notably higher than that in other parts of China (Fig. 5A). However, uniform high heat stress is projected for the entire eastern China by the end of the century under both the SSP2-4.5 and SSP5-8.5 scenarios (Fig. 5B, C).”
- For your another concern about the similarities of the findings with other previous studies, we agree that a large number of existing studies have explored variations of extreme heat (Ding et al., 2010; Hu et al., 2017, 2023; Li and Zha, 2020; Li and Amatus, 2020; Sun et al., 2014; Wang and Yan, 2021). They attributed the increasing trends of historical extreme temperature and heat waves in China since 1960s to greenhouse gas emissions (Dong et al., 2018; Li et al., 2022a; Ma et al., 2017; Sun et al., 2014, 2022;), to anthropogenic aerosol emissions (Chen et al., 2019), to advection from tropical regions (Freychet et al., 2017), and to urbanization (Wu et al., 2021; Yang et al., 2017). Future exposure to extreme temperatures in China were also explored extensively (Huang et al., 2018; Liu et al., 2018; Zhang et al., 2021; Li et al., 2018; Yao et al., 2023; Zhang et al., 2020; Zhu et al., 2017; Leng et al., 2016).
- However, existing studies primarily focused on heat extremes defined solely by temperature, such as extreme high temperature or heat wave events. Only a few studies investigated historical variation (Wang and Sun, 2022; Li et al., 2017) and future projections of T_w across China (Li et al., 2022; Chen et al., 2022). China's extreme T_w has undergone rapid intensification since 1960s (Wang and Sun, 2022; Li et al., 2017). A global study published in Nature Climate Change (Li et al., 2018) emphasized that apparent temperatures that consider humidity would increase faster than air temperature under high greenhouse gas emissions.
- Yet humidity may change differently across regions, as found by our studies. We emphasized here that external forcings induced changes in moisture transport and local evaporation, which exhibited north-south differences in China. As a result, China is likely to experience a more widely-spread and uniformly elevated heat stress in the future. The conclusions were supported by both historical data and future projections. These points have not been reported in any previous studies.
- To address your concern, we re-organized introduction and discussion to emphasize

- the difference and novelty of this study.
- “China's T_w has undergone rapid intensification since 1960s, primarily attributed to human induced climate change, exerting substantial heat stress on its vast population. However, both rising and falling trends in atmospheric moisture were identified across China. For example, notable decreases in atmospheric moisture were found in South China during 1961–2014 and Southwest China during 1979–2013, and increasing tendencies were confirmed in the Yangtze River basin during 1961-2005. This indicates that T_w is likely to change differently across regions under climate change, which has not been fully understood. Given the significance of heat stress with respect to human health and food security, a better understanding of how historical and future T_w evolves in different regions and the key driving factors would better assist mitigation and adaptation of heat stress, particularly for populous country like China.”
 - “In this study, we revealed the spatially differentiated historical T_w changes in China, characterized by a faster rise of T_w and a more pronounced aggravation of heat stress in northern China. We found the dominant role of changes in E_a and attributed such heterogeneity to declining E_a across most of the southern areas, which was associated with faster warming in high-latitude regions of East Asia. A faster warming of high-latitudes of East Asia regulates atmospheric features, leading to extended impacts of the SAH and WPSH over southern China and suppressed moisture transport under global warming.”

Reference:

- Chen, H., He, W., Sun, J. and Chen, L., 2022. Increases of extreme heat-humidity days endanger future populations living in China. *Environmental Research Letters*, 17(6), p.064013.
- Chen, W., Dong, B., Wilcox, L., Luo, F., Dunstone, N. and Highwood, E.J., 2019. Attribution of recent trends in temperature extremes over China: role of changes in anthropogenic aerosol emissions over Asia. *Journal of Climate*, 32(21), pp.7539-7560.
- Ding, T., Qian, W. and Yan, Z., 2010. Changes in hot days and heat waves in China during 1961–2007. *International Journal of Climatology*, 30(10), pp.1452-1462.
- Dong, S., Sun, Y., Aguilar, E., Zhang, X., Peterson, T.C., Song, L. and Zhang, Y., 2018. Observed changes in temperature extremes over Asia and their attribution. *Climate Dynamics*, 51, pp.339-353.
- Freychet, N., Tett, S., Wang, J. and Hegerl, G., 2017. Summer heat waves over eastern China: Dynamical processes and trend attribution. *Environmental Research Letters*, 12(2), p.024015.
- He, W. and Chen, H., 2023. More extreme-heat occurrences related to humidity in China. *Atmospheric and Oceanic Science Letters*, p.100391.
- Hu, L., Huang, G. and Qu, X., 2017. Spatial and temporal features of summer extreme temperature over China during 1960–2013. *Theoretical and applied climatology*, 128, pp.821-833.
- Hu, X., Cao, J., Qian, Y., Zhou, W. and Zheng, Z., 2024. Extreme heat events in mainland China from 1981 to 2015: Spatial patterns, temporal trends, and urbanization impacts. *Sustainable Cities and Society*, 100, p.104999.
- Huang, D., Zhang, L., Gao, G. and Sun, S., 2018. Projected changes in population exposure to extreme heat in China under a RCP8.5 scenario. *Journal of Geographical Sciences*, 28, pp.1371-1384.
- Leng, G., Tang, Q., Huang, S. and Zhang, X., 2016. Extreme hot summers in China in the CMIP5 climate models. *Climatic Change*, 135, pp.669-681.
- Li, C., Zhang, X., Zwiers, F., Fang, Y. and Michalak, A.M., 2017. Recent very hot summers in Northern Hemispheric land areas measured by wet bulb globe temperature will be the norm

- within 20 years. *Earth's Future*, 5(12), pp.1203-1216.
- Li, J., Chen, Y.D., Gan, T.Y. and Lau, N.C., 2018. Elevated increases in human-perceived temperature under climate warming. *Nature Climate Change*, 8(1), pp.43-47.
 - Li, K. and Amatus, G., 2020. Spatiotemporal changes of heat waves and extreme temperatures in the main cities of China from 1955 to 2014. *Natural Hazards and Earth System Sciences*, 20(7), pp.1889-1901.
 - Li, L. and Zha, Y., 2020. Population exposure to extreme heat in China: Frequency, intensity, duration and temporal trends. *Sustainable Cities and Society*, 60, p.102282.
 - Li, L., Yao, N., Li, Y., Li Liu, D., Wang, B. and Ayantobo, O.O., 2019. Future projections of extreme temperature events in different sub-regions of China. *Atmospheric research*, 217, pp.150-164.
 - Li, W., Hao, X., Wang, L., Li, Y., Li, J., Li, H. and Han, T., 2022. Detection and attribution of changes in thermal discomfort over China during 1961–2014 and future projections. *Advances in Atmospheric Sciences*, 39(3), pp.456-470.
 - Li, W., Jiang, Z., Li, L.Z., Luo, J.J. and Zhai, P., 2022. Detection and attribution of changes in summer compound hot and dry events over northeastern China with CMIP6 models. *Journal of Meteorological Research*, 36(1), pp.37-48.
 - Liu, X., Tang, Q., Zhang, X. and Sun, S., 2018. Projected changes in extreme high temperature and heat stress in China. *Journal of Meteorological Research*, 32(3), pp.351-366.
 - Ma, S., Zhou, T., Stone, D.A., Angélil, O. and Shiogama, H., 2017. Attribution of the July–August 2013 heat event in Central and Eastern China to anthropogenic greenhouse gas emissions. *Environmental Research Letters*, 12(5), p.054020.
 - Sun, Y., Zhang, X., Ding, Y., Chen, D., Qin, D. and Zhai, P., 2022. Understanding human influence on climate change in China. *National science review*, 9(3), p.nwab113.
 - Sun, Y., Zhang, X., Zwiers, F.W., Song, L., Wan, H., Hu, T., Yin, H. and Ren, G., 2014. Rapid increase in the risk of extreme summer heat in Eastern China. *Nature Climate Change*, 4(12), pp.1082-1085.
 - Wang, D. and Sun, Y., 2021. Long-term changes in summer extreme wet bulb globe temperature over China. *Journal of Meteorological Research*, 35, pp.975-986.
 - Wang, J. and Yan, Z., 2021. Rapid rises in the magnitude and risk of extreme regional heat wave events in China. *Weather and Climate Extremes*, 34, p.100379.
 - Wu, S., Wang, P., Tong, X., Tian, H., Zhao, Y. and Luo, M., 2021. Urbanization-driven increases in summertime compound heat extremes across China. *Science of the Total Environment*, 799, p.149166.
 - Yang, X., Ruby Leung, L., Zhao, N., Zhao, C., Qian, Y., Hu, K., Liu, X. and Chen, B., 2017. Contribution of urbanization to the increase of extreme heat events in an urban agglomeration in east China. *Geophysical Research Letters*, 44(13), pp.6940-6950.
 - Yao, Y., Zhang, W. and Kirtman, B., 2023. Increasing impacts of summer extreme precipitation and heatwaves in eastern China. *Climatic Change*, 176(10), p.131.
 - Zhang, G., Zeng, G., Yang, X. and Jiang, Z., 2021. Future changes in extreme high temperature over China at 1.5 C–5 C global warming based on CMIP6 simulations. *Advances in Atmospheric Sciences*, 38, pp.253-267.
 - Zhang, G.W., Gang, Z.E.N.G., Iyakaremye, V. and Qing-Long, Y.O.U., 2020. Regional changes in extreme heat events in China under stabilized 1.5 C and 2.0 C global warming. *Advances in Climate Change Research*, 11(3), pp.198-209.
 - Zhu, J., Huang, G., Wang, X. and Cheng, G., 2017. Investigation of changes in extreme temperature and humidity over China through a dynamical downscaling approach. *Earth's Future*, 5(11), pp.1136-1155.

More specific comments can be found in the following.

1. Line 61: Is China the second-largest population now?

Reply:

- Yes, India overtook China as the country with the largest population in the world in 2023, according to information from several sources.
- <https://www.statista.com/statistics/262879/countries-with-the-largest-population/>
- United Nations also reported this last year: “In April 2023, India’s population is expected to reach 1,425,775,850 people, matching and then surpassing the population of mainland China (Figure 1).”
- <https://www.un.org/en/desa/india-overtake-china-world-most-populous-country-april-2023-united-nations-projects>
- The statistics may need to be further confirmed later, so we revised the sentence to avoid any concern on the accuracy.
- “exerting substantial heat stress on its vast population”.

2. Line 83: “increasing” may be better than “enhancement”

Reply:

- Done as suggested.

3. Line 90: The regions’ names are needed to be labeled in Fig. 1A

Reply:

- Done as suggested.

Fig. R1.1. Spatiotemporal variations of wet bulb temperature (T_w) in China. (A) Spatial distribution of average summertime T_w during the period from 1979 to 2018. Black squares represent four key agglomerations: Beijing-Tianjin-Hebei (BTH, 38°N-41°N, 115°E-120°E), Yangtze River Delta (YRD, 29°N-33°N, 118°E-123°E), Sichuan Basin (SCB, 29°N-32°N, 103°E-107°E) and Pearl River Delta (PRD, 21°N-23°N, 112°E-115°E). (B) Spatial distribution of T_w trend during the period from 1979 to 2018. Only sites with significant trend ($P < 0.05$) are displayed. Green dashed line indicates the latitude of 33°N. (C) Time series of average T_w anomaly of northern and southern stations during the period from 1979 to 2018. (D) Time series of average T_w of BTH, YRD, SCB and PRD during the period from 1979 to 2018.

4. Line 97:- Here, the authors have investigated the sensitivity of T_w to T and E_a . They indicated that the changes of T_w in northern China and southern China are mainly caused by changes in E_a . How do you get this conclusion? From Fig. 2, I think the role of temperature seems to be much greater than E_a . If the conclusion you obtained is ok, more evident are needed, for example, how many contribution from T or E_a should be clear.

Reply:

- Sorry for the vagueness. We investigated changes of T_w when changed T or E_a only

but keep the value of the other factor at year 1979 (Fig. R1.2). When E_a changes with time but T is fixed at year 1979, T_w shows more similar variations with observed patterns in southern China, displaying negative trends (Fig. R1.1B, Fig. R1.2D). This suggests that the different trends observed in Southern China is more likely to be caused by E_a changes (Fig. R1.2A). While in northern China, T dominates still changes of T_w .

- To avoid confusion, we changed all related expressions in the revised manuscript.
- “Changes in T_w when either T or E_a varied but the other factor fixed at year 1979 highlight that E_a changes are responsible for the observed different T_w trends in southern China, while T dominates still changes of T_w in northern China (Fig. S4), but T dominates still changes of T_w in northern China.”

Fig. R1.2. Air temperature (T), water vapor (E_a) and wet bulb temperature (T_w) variations. Spatial distribution of T (A) and E_a (B) variations during the period from 1979 to 2018. Only sites with significant trend ($P < 0.05$) are displayed. Spatial distribution of T_w trend under only T changes (C) and only E_a changes (D) conditions during the period from 1979 to 2018. Only sites with significant trend ($P < 0.05$) are displayed.

5. Line 130-: How to identify the time of the land use conversion for each station.

Reply:

- We used gridded land use dataset obtained from National Tibetan Plateau Data

Center (<https://data.tpcd.ac.cn>) to identify the time of the land use conversion. The data set included seven periods: the end of 1980s, 1990, 1995, 2000, 2005, 2010 and 2015. We determined conversion time as the first time that the grid of the station location changed to urban land.

- For example, if land use type of one station was nonurban in 1995 but became urban land use in 2000, we determined the conversion time of this station to be 2000.
- To address your concern and make it clear to the readers, we added the following sentence in the revised manuscript.
- “We determined conversion time as the first time that the grid of the station location changed to urban land use.”

6. In this study, the authors have evaluated contribution of GHG, AER, NAT, and LU to the change in T_w . It is ok. I want to know whether these signals can be detected for the observed change in T_w in China. But I have not found any information from the current MS.

Reply:

- Thank you for raising this question. Observed changes in T_w in China could be affected by all these factors. However, the effects of single forcing cannot be directly detected.
- To evaluate contributions of GHG, AER, NAT and LU to changes in T_w from observations, we employed empirical decomposition method (EMD) to separate internal natural variability and external anthropogenic forcing. The residual obtained from the EMD analysis can be considered as the impacts of anthropogenic emissions, urbanization and land use change (Loehle and Scafetta, 2011).
- We found the natural variability exhibited insignificant trends in T_w at most sites in China (Fig. R1.3A). However, external forcings emerge as the dominant factor driving T_w changes during the period of 1979-2018 (Fig. R1.3B). This is evident from the consistent spatial patterns observed, which align with our original findings.
- We further employed empirical orthogonal function (EOF) decomposition on derived T_w values resulting from external forcings to identify the contributions of each anthropogenic factor including GHG, AER and LU (Fig. R1.4). EOF1 and EOF2 can be considered as the contributions of GHG and AER, respectively, as they exhibit similar spatial patterns as our attribution analysis (Fig. R1.4A, B, Fig. R1.5C, D). They account for 68.6% and 23.8% of the total variations, respectively, which closely aligns with the results obtained from our attribution analysis (GHG: ~63%, AER: ~24%). The third mode of EOF shows a generally decreasing trend of T_w , which can be attributed to the impacts of urbanization and land use changes (Fig. R1.4C).
- Following your suggestion, we added these results in the revised manuscript:
- “We also employed the empirical decomposition method (EMD) on T_w variations first to separate internal natural variability and external anthropogenic forcing, and then applied empirical orthogonal function (EOF) decomposition on external signal. Similar contributions of GHG (~68.6%) and AER (~23.8%) to T_w variations were found.”

Fig. R1.3. Empirical decomposition analysis on T_w . Spatial distribution of T_w trends caused by natural variabilities (A) and external forcings (B) during the period of 1979-2018.

Fig. R1.4. Empirical orthogonal function (EOF) decomposition of T_w . Spatial patterns of (A) EOF1, (B) EOF2, and (C) EOF3.

Fig. R1.5. Observed and simulated variations of T_w . (A) Spatial distribution of observed T_w variations during the period of 1979-2014. Only sites with significant trend ($P < 0.05$) are displayed. (B) Spatial distribution of simulated T_w variations under all-forcing conditions during the period of 1979-2014. Black dots denote areas with significant trend ($P < 0.05$). Spatial distribution of simulated T_w variations under GHG-only (C), aerosols-only (D), natural-only (E) and land use-only (F) forcing conditions during the period of 1979-2014. Black dots denote areas with significant trend ($P < 0.05$).

Reference:

- Gillett, N. P. et al. The detection and attribution model intercomparison project (DAMIP v1.0) contribution to CMIP6. *Geoscientific Model Development* 9, 3685-3697 (2016).
- Lawrence, D. M. et al. The Land Use Model Intercomparison Project (LUMIP) contribution to CMIP6: rationale and experimental design. *Geoscientific Model Development* 9, 2973-2998

(2016).

- Loehle, C. and Scafetta, N., 2012. Climate change attribution using empirical decomposition of climatic data. arXiv preprint arXiv:1206.5845.

7. Figure S8: The GHG presents a uniformly influence on T_w across China. But there is almost reversing impact of AER over northern and southern China. Why?

Reply:

- Impacts of aerosols on T_w variations in China are determined by changes of anthropogenic aerosol emission across China during the period of 1979-2014. Anthropogenic aerosol emissions mainly show increasing trend over regions to the south of Inner Mongolia (Fig. R1.6), leading to a decreased surface air temperature over southern China (Fig. R1.7A). As a result, we observed cyclone anomaly at lower levels over that region, accelerating moisture transported to northern China leading to faster increase in E_a (Fig. R1.7B, C, D). Under the combined effects of T and E_a , there is reversing impacts of AER over northern and southern China.

Fig. R1.6. Anthropogenic aerosol emission variations. Spatial distribution of surface anthropogenic aerosol emission variations from the CMIP6 during the period from 1979 to 2014. Black dots denote areas with significant trend ($P < 0.05$).

Fig. R1.7. T, E_a and atmospheric circulation changes. Spatial distribution of T (A) and E_a (B) variations from aerosols-only simulations during the period from 1979 to 2014. Black dots denote areas with significant trend ($P < 0.05$). Spatial distribution of differences in average summertime geopotential height and atmospheric circulation on 700 hPa (C) and 850 hPa (D) between the period of 1979-1996 and 1997-2014.

8. Line 143: Note the difference between “predict” and “project”

Reply:

- We changed it to “projected”.

9. “SSP245” and “SSP585” should be “SSP2-4.5” and “SSP5-8.5”

Reply:

- We changed them across the manuscript.

10. Line 151: Why the rise in T_w in northern China shows much greater than that in southern China in the future?

Reply:

- As elaborated in the manuscript, future increases in T display a less distinction between northern and southern regions (Fig. R1.8A, B, Fig. R1.9B). In contrast, there are notable spatial difference in E_a changes, characterized by a more pronounced elevation in northern China (~ 2 hPa higher, Fig. R1.8C, D, Fig. R1.9C).
- This phenomenon can be attributed to anticipated shifts in atmospheric systems. Under a warming climate, both SAH and WPSH exhibit substantial intensification, with a noticeable increase in central pressure (Fig. R1.10). As a result, moisture transport from both the Indian Ocean and the Pacific Ocean to southern China is reduced, different from the enhanced moisture transport to northern China (Fig. R1.11). We have included a more comprehensive explanation in the revised manuscript.
- “In accordance with historical trends, simulated increases in T display a subtle absence of distinction between northern and southern regions (Fig. S18A, B). In contrast, there are notable spatial differences in E_a changes, characterized by a more pronounced elevation in northern China (Fig. S18C, D). This phenomenon can be attributed to anticipated shifts in atmospheric systems (Fig. S19, Fig. S20). Under a warming climate, both SAH and WPSH exhibit substantial intensification, with a notable increase in central pressure (Fig. S19), resulting in anticyclone enhancements over the Bay of Bangladesh and South China Sea at lower layers (Fig. S21E, F, G, H). As a result, influenced by the air flow at the ridge of anticyclone, more water vapor is transported to the Pacific Ocean instead of to southern China, different from the enhanced moisture transport from the Pacific Ocean to northern China (Fig. S21). Under both the SSP2-4.5 and SSP5-8.5 scenarios, the increase in T_w resulting from changes in E_a is notably more substantial than that by T across most regions of China (Fig. 6). The impact of elevated E_a contributes roughly twice of the influence of T to increased T_w , particularly in humid and semi-humid areas.”

Fig. R1.8. Future changes of T and E_a. Spatial distribution of future changes of T under the SSP2-4.5 (A) and SSP5-8.5 (B) scenarios from WRF-Chem simulations. Spatial distribution of future changes of E_a under the SSP2-4.5 (C) and SSP5-8.5 (D) scenarios from WRF-Chem simulations.

Fig. R1.9. Future changes of T_w, T and E_a. Future changes of T_w (A), T (B) and E_a (C) in northern and southern China under the SSP2-4.5 and SSP5-8.5 (B) scenarios from WRF-Chem simulations.

Fig. R1.10. Historical and future changes of atmospheric circulations. Spatial distribution of average summertime geopotential height and circulation on 100 hPa over the Hist period (2010–2014) (A) and future period (1996–2100) under the SSP2-4.5 (C) and SSP5-8.5 (E) scenarios from bias corrected CMIP6 global dataset. Spatial distribution of average summertime geopotential height on 500 hPa over the Hist period (2010–2014) (B) and future period (1996–2100) under the SSP2-4.5 (D) and SSP5-8.5 (F) scenarios from bias corrected CMIP6 global dataset.

Fig. R1.11. Future changes of atmospheric circulations. Spatial distribution of future changes of summertime geopotential height and circulation at 100 hPa (A), 500 hPa (C), 700 hPa (E) and 850 hPa (G) under the SSP2-4.5 scenarios from bias corrected CMIP6 global dataset. Spatial distribution of future changes of summertime geopotential height and circulation at 100 hPa (B), 500 hPa (D), 700 hPa (F) and 850 hPa (H) under the SSP5-8.5 scenarios from bias corrected CMIP6 global dataset.

11. Line 155: Is “northern China” ok?

Reply:

- Yes, both North China and northern China are grammatically correct.

12. Figure S10: The change patterns are much better than the climatic means.

Reply:

- We apologize for using the improper caption of Figure S10 that may have caused some ambiguity. This figure does not display change patterns but instead illustrates the spatial distribution of average summertime geopotential height and circulation at 100 hPa and 500 hPa over the Hist period (2010-2014) and future period (1996-2100) under the SSP2-4.5 and SSP5-8.5 scenarios from bias corrected CMIP6 global dataset.
- We revised the caption to “Historical and future atmospheric circulations” (see Fig. R1.12).
- Figure S11 in the original manuscript (Fig. R1.13 here, Fig. S19 in the revised manuscript) shows change patterns.

Fig. R1.12. Historical and future atmospheric circulations. Spatial distribution of average summertime geopotential height and circulation on 100 hPa over the Hist period (2010-2014) (A) and future period (1996-2100) under the SSP245 (C) and SSP585 (E) scenarios from bias corrected CMIP6 global dataset. Spatial distribution of average summertime geopotential height on 500 hPa over the Hist period (2010-2014) (B) and

future period (1996-2100) under the SSP245 (D) and SSP585 (F) scenarios from bias corrected CMIP6 global dataset.

Fig. R1.13. Future changes of atmospheric circulations. Spatial distribution of future changes of summertime geopotential height and circulation at 100 hPa under the SSP2-4.5 (A) and SSP5-8.5 (B) scenarios from bias corrected CMIP6 global dataset. Spatial

distribution of future changes of summertime geopotential height and circulation at 500 hPa under the SSP2-4.5 (C) and SSP5-8.5 (D) scenarios from bias corrected CMIP6 global dataset. Spatial distribution of future changes of summertime geopotential height and circulation at 700 hPa under the SSP2-4.5 (E) and SSP5-8.5 (F) scenarios from bias corrected CMIP6 global dataset. Spatial distribution of future changes of summertime geopotential height and circulation at 850 hPa under the SSP2-4.5 (G) and SSP5-8.5 (H) scenarios from bias corrected CMIP6 global dataset.

13. Line 167: This description is not correct and there have been many such works.

Reply:

- We agree that there have been some studies focusing on historical and future T_w variations and their dominant drivers in China (Ning et al., 2021; Wang and Sun, 2022; Li et al., 2017, 2020; Chen et al., 2022; Wang et al., 2019). Yet humidity may change differently across regions, as found by our study. We emphasized here that external forcings induced changes in moisture transport and local evaporation, which exhibited north-south differences in China. As a result, China is likely to experience a more widely-spread and uniformly elevated heat stress in the future. The conclusions were supported by both historical data and future projections. These points have not been reported in any previous studies.
- To address your concern, we revised the sentence to the following:
- “Yet spatial heterogeneity of historical and future T_w variations in China and the underlying reasons have not been well understood.”

Reference:

- Chen, H., He, W., Sun, J. and Chen, L., 2022. Increases of extreme heat-humidity days endanger future populations living in China. *Environmental Research Letters*, 17(6), p.064013.
- Li, C., Zhang, X., Zwiers, F., Fang, Y. and Michalak, A.M., 2017. Recent very hot summers in Northern Hemispheric land areas measured by wet bulb globe temperature will be the norm within 20 years. *Earth's Future*, 5(12), pp.1203-1216.
- Ning, G., Luo, M., Wang, S., Liu, Z., Wang, P. and Yang, Y., 2022. Dominant modes of summer wet bulb temperature in China. *Climate Dynamics*, pp.1-16.
- Wang, D. and Sun, Y., 2021. Long-term changes in summer extreme wet bulb globe temperature over China. *Journal of Meteorological Research*, 35, pp.975-986.
- Wang, P., Leung, L.R., Lu, J., Song, F. and Tang, J., 2019. Extreme wet - bulb temperatures in China: the significant role of moisture. *Journal of Geophysical Research: Atmospheres*, 124(22), pp.11944-11960.
- Li, C., Sun, Y., Zwiers, F., Wang, D., Zhang, X., Chen, G. and Wu, H., 2020. Rapid warming in summer wet bulb globe temperature in China with human-induced climate change. *Journal of Climate*, 33(13), pp.5697-5711.

14. Line 209: There are many calculations for T_w . Their difference may be need for discussion

Reply:

- We agree that there are many calculations for T_w , but almost all calculations use surface air temperature, humidity, and pressure as inputs (Davies-Jones, 2008;

Krakauer et al., 2020). As the dataset used in this study lacks pressure measurements, we used an empirical computation formula (Stull, 2011), which use atmospheric surface temperature and RH (contains both humidity and pressure information), both being available for station dataset.

- The Stull formulation is valid for RH between 5% and 99% and for temperatures from -20 to 50 °C (Stull, 2011), and the climate of our study area falls perfectly within this requirement. This method has been well adopted by T_w related studies in China (Li et al., 2017; Ning et al., 2022; Xiao et al., 2022; Wang and Sun, 2021; Freychet et al., 2020). We highlighted this in Materials and Methods.
- “This formulation is applicable for relative humidity (RH) ranging from 5% to 99% and temperatures ranging from -20 to 50 °C, which are reasonable ranges for this study.”
- To address your concern, we also computed the average T_w and T_w variation using three most widely adopted methods proposed by Stull (2010) (this study), Davies-Jones (2008), and Krakauer et al. (2020) based on ERA5 data.
- In Stull’s method, T_w was calculated using equation with inputs of air temperature and relative humidity (1):

$$T_w = T \cdot \text{atan} \left[0.151977(RH + 8.313659)^{\frac{1}{2}} \right] + \text{atan}(T + RH) - \text{atan}(RH - 1.676331) + 0.00391838(RH)^{\frac{3}{2}} \cdot \text{atan}(0.023101 \cdot RH) - 4.686035 \quad (1)$$

- Obtaining T_w using Davies-Jones’s method was to solve the following equation:

$$T_E = T_w \left[1 - \frac{e_s(T_w)}{p_0 \pi^\lambda} \right]^{-\nu} \pi^{k_3 r_s(T_w, \pi)} e^{G(T_w, \pi)} \quad (2)$$

where

$$G(T_w, \pi) = \left(\frac{k_0}{T_w} - k_1 \right) [r_s(T_w, \pi) + k_2 r_s^2(T_w, \pi)] \quad (3)$$

- Solving equation (2) requires surface temperature, relative humidity and pressure as inputs. More detailed calculation steps can be found in the referred paper (Davies-Jones, 2008).
- In Krakauer’s method, T_w was calculated using equation (4-7) with inputs of surface temperature, relative humidity, and pressure.

$$h_{sat, T_w} = h + (r_{sat} - r)h_{L, T_w} \quad (4)$$

$$(r_{sat} - r)h_{L, T_w} = (C_{p,a} - rC_{p,v})(T - T_w) \quad (5)$$

$$r_{sat} = \frac{P_{sat, T_w}}{p - P_{sat, T_w}} \quad (6)$$

$$P_{sat} = (1.56 \times 10^{11} Pa) e^{-\frac{5.42 \times 10^3 K}{T}} \quad (7)$$

- The results obtained from the first two methods exhibit a comparable spatial distribution of average and variation in T_w (Fig. R1.14A, B, D, E). While some discrepancies are observed in the results obtained from Krakauer’s method, we still observe a decrease in T_w from the southeastern coastal areas towards the inland regions (Fig. R1.14C). Additionally, T_w in the northern regions of China shows a faster increasing rate compared to the southern regions (Fig. R1.14F).
- Similar spatial distribution of station-based T_w that decreased from the southeastern coastal areas toward inland regions was also observed in another study (Wang et al.,

2019) in China using the calculation method from Davies-Jones (2008) (Fig. R1.15). We have included this discussion in Materials and Methods.

- “ T_w calculated with other methods^{68,69} showed comparable spatial distributions of average and variation over the period of 1979-2018 (Fig. S22).”

Fig. R1.14. Historical average and variations of T_w . Spatial distribution of average summertime T_w calculated following method from Stull (2010) (adopted in this study) (A), Davies-Jones (2008) (B), and Krakauer et al. (2020) (C) based on ERA5 data. Spatial distribution of trend of summertime T_w calculated following method from Stull (2010) (D), Davies-Jones (2008) (E), and Krakauer et al. (2020) (F) based on ERA5 data. Black dots denote areas with significant trend ($P < 0.05$).

Fig. R1.15. Spatial distribution of the averaged daily maximum TW (°C) during May to September over the period 1960–2015 from Wang et al. (2019).

Reference:

- Davies-Jones, R., 2008. An efficient and accurate method for computing the wet-bulb temperature along pseudoadiabats. *Monthly Weather Review*, 136(7), pp.2764-2785.
- Freychet, N., Tett, S.F.B., Yan, Z. and Li, Z., 2020. Underestimated change of wet - bulb temperatures over East and South China. *Geophysical Research Letters*, 47(3), p.e2019GL086140.
- Krakauer, N.Y., Cook, B.I. and Puma, M.J., 2020. Effect of irrigation on humid heat extremes. *Environmental Research Letters*, 15(9), p.094010.
- Li, C., Zhang, X., Zwiers, F., Fang, Y. and Michalak, A.M., 2017. Recent very hot summers in Northern Hemispheric land areas measured by wet bulb globe temperature will be the norm within 20 years. *Earth's Future*, 5(12), pp.1203-1216.
- Ning, G., Luo, M., Wang, S., Liu, Z., Wang, P. and Yang, Y., 2022. Dominant modes of summer wet bulb temperature in China. *Climate Dynamics*, pp.1-16.
- Wang, D. and Sun, Y., 2021. Long-term changes in summer extreme wet bulb globe temperature over China. *Journal of Meteorological Research*, 35, pp.975-986.
- Wang, P., Leung, L.R., Lu, J., Song, F. and Tang, J., 2019. Extreme wet - bulb temperatures in China: the significant role of moisture. *Journal of Geophysical Research: Atmospheres*, 124(22), pp.11944-11960.
- Xiao, X., Xu, Y., Zhang, X., Wang, F., Lu, X., Cai, Z., Brasseur, G. and Gao, M., 2022. Amplified upward trend of the joint occurrences of heat and ozone extremes in China over 2013–20. *Bulletin of the American Meteorological Society*, 103(5), pp.E1330-E1342.

15. Line 212: “due point”?

Reply:

- We changed it to “dew point”.

16. Line 243: *In general, more runs are much better for such analysis*

Reply:

- Thanks for your suggestion. In the revised manuscript, we used an ensemble mean of all runs contributed by each single model instead of the first run. We calculated the ensemble average of runs for each model and then calculated the ensemble mean of all models to ensure equal weighting for each model in this analysis. We clarified this in Methods and Materials.
- “We calculated the ensemble mean of all runs contributed by each single model and then calculated the average of all models to ensure equal weighting for each model in this analysis.”
- There are only minor changes in spatial patterns (Fig. R1.12 vs Fig. R1.13) and attribution analysis (Fig. R1.14 vs Fig. R1.15, Fig. R1.16 vs Fig. R1.17), which means that these changes do not affect our previous findings.

Fig. R1.16. Observed and simulated variations of T_w (revised version). (A) Spatial distribution of observed T_w variations during the period of 1979-2014. Only sites with significant trend ($P < 0.05$) are displayed. (B) Spatial distribution of simulated T_w variations under all-forcing conditions during the period of 1979-2014. Black dots denote areas with significant trend ($P < 0.05$). Spatial distribution of simulated T_w variations under GHG-only (C), aerosols-only (D), natural-only (E) and land use-only (F) forcing conditions during the period of 1979-2014. Black dots denote areas with significant trend ($P < 0.05$).

Fig. R1.17. Observed and simulated variations of T_w (original version). (A) Spatial distribution of observed T_w variations during the period of 1979-2014. Only sites with significant trend ($P < 0.05$) are displayed. (B) Spatial distribution of simulated T_w variations under all-forcing conditions during the period of 1979-2014. Black dots denote areas with significant trend ($P < 0.05$). Spatial distribution of simulated T_w variations under GHG-only (C), aerosols-only (D), natural-only (E) and land use-only (F) forcing conditions during the period of 1979-2014. Black dots denote areas with significant trend ($P < 0.05$).

Fig. R1.18. Attributable variations of T_w from different contributors (revised version). Contributions of individual forcing to variations of T_w caused by all forcings. Yellow lines are median values, box chart values denote median values minus standard deviation, 25% quantile, 75% quantile, and the median value plus standard deviation from bottom to top, respectively.

Fig. R1.19. Attributable variations of T_w from different contributors (original version). Contributions of individual forcing to variations of T_w caused by all forcings. Yellow lines are median values, box chart values denote median values minus standard deviation, 25% quantile, 75% quantile, and the median value plus standard deviation from bottom to top, respectively.

Fig. R1.20. Contributions of individual forcing to variations of T_w (revised version). Attributable variations of T_w from GHG-only (A), aerosols-only (B), natural-only (C) and land use-only (D) forcing conditions during the period of 1979-2014.

Fig. R1.21. Contributions of individual forcing to variations of T_w (original version). Attributable variations of T_w from GHG-only (A), aerosols-only (B), natural-only (C) and land use-only (D) forcing conditions during the period of 1979-2014.

17. Line 244: Which method is used for the interpolation? Is the topographical effect considered for the temperature interpolation?

Reply:

- We used liner interpolation, and we clarified it in the revised manuscript.
- “The model outputs were interpolated to the resolution of $2^\circ \times 2^\circ$ using liner interpolation, and the details of selected models are shown in Table S1.”
- We did not consider topographical effect because the topographical effect could be neglected under such coarse resolutions (hundreds of kilometers) of both original (see resolution in Table R1.1) and target grids ($2^\circ \times 2^\circ$).
- We also compared the summertime average T_w over the period of 1979-2014 before and after interpolation, using 12 experiments from CMIP6 that used in this study (Fig. R1.22). Our analyses reveal no discernible difference in the spatial distribution of T_w before and after interpolation. T_w at the corresponding positions of the 826 observation sites adopted in this study also exhibit small differences before and after interpolation, characterized by high correlation coefficients and low root mean squared errors (Fig. R1.23). These results suggest that neglecting the

topographical effect has minimal impact on the findings presented in this study.
 Table R1.1 A list of CMIP6 models.

Model Name	Developer	Resolution (lat × lon)
ACCESS-CM2	Commonwealth Scientific and Industrial Research, Australia	1.25° × 1.875°
ACCESS-ESM1-5	Commonwealth Scientific and Industrial Research, Australia	1.25° × 1.875°
CESM2	National Center for Atmospheric Research, USA	0.9375° × 1.25°
CanESM5	Canadian Centre for Climate, Canada	2.815° × 2.815°
GFDL-ESM4	Geophysical Fluid Dynamics Laboratory, USA	1° × 1.25°
GISS-E2-1-G	NASA Goddard Institute for Space Studies, USA	2° × 2.5°
IPSL-CM6A-LR	Institut Pierre-Simon Laplace, France	1.25° × 2.5°
MIROC6	Atmosphere and Ocean Research Institute, Japan	1.40° × 1.40°
MRI-ESM2-0	Meteorological Research Institute, Japan	1.125° × 1.125°
FGOALS-g3	Institute of Atmospheric Physics, China	2.25° × 2°
CMCC-ESM2	Centro Euro-Mediterraneo sui Cambiamenti Climatici, Italy	1.875° × 1.875°
MPI-ESM1-2	Max Planck Institute for Meteorology, Germany	1.241° × 1.875°

Fig. R1.22 Spatial distribution of average T_w . Spatial distribution of summertime average T_w over 1979-2014 from 12 CMIP6 experiments used in this study before (A1-L1) and after (A2-L2) interpolation.

Fig. R1.23 Density scatterplots of T_w . Density scatterplots of T_w before and after interpolation at the corresponding positions of the 826 observation sites over 1979-2014 from 12 CMIP6 experiments used in this study.

18. Line 246:- I wonder why the authors used downscaling simulations for the future projection. Additionally, the models are used for the drivers of the downscaling are different from the early section of attribution analyses, which seem to be much arbitrary for the analysis in this study and the conclusions may be inconsistent across this MS.

Reply:

- As mentioned in replies above, the major objective of this study is to understand how T_w evolved in the past and will change in the future. We learned from historical observed variations of T_w and attribution analysis. Future projections were used to confirm the relationship and mechanism found in historical records, and to tell the public what would happen in the future.
- We adopted WRF-Chem downscaling simulations for the future projection because we wanted to obtain a high resolution and accurate projection of future shifts of T_w across the whole China under the SSP2-4.5 and SSP5-8.5 scenarios. Spatial resolution of horizontal grids from original CMIP6 experiments are mostly larger than 1° (about one hundred kilometers) which is insufficient for precise regional

- heat stress prediction and effective mitigation strategy development.
- The historical simulations were forced by FNL reanalysis dataset. To keep consistent, we adopted bias corrected CMIP6 global dataset which has been adjusted using the ERA5 reanalysis dataset (Xu et al., 2021) instead of same experiments adopted in the attribution analyses as large-scale forcing for the future simulations. ERA5 and FNL are two similar reanalysis datasets that assimilate various types of observations such as surface observations, radiosonde data, satellite data, buoy data, and aircraft data. They provide consistent large-scale atmospheric forcing for WRF-Chem simulations (Fig. R1.24). In addition, using bias corrected CMIP6 global dataset can provide more accurate future projection because dynamical downscaling simulations are often degraded by biases in the large-scale forcing itself (Xu et al., 2021; Wu et al., 2022; Maraun, 2016).
 - From historical observations and attribution analysis, we concluded that atmospheric circulation changes under global warming leads to water vapor dominated south-to-north spatially heterogeneous trends of T_w in China. We observed widespread and uniform elevated water vapor and thermal stress at the end of this century because of a continuous warming climate under future high emission scenarios. Past and future results corroborate each other, proving that the conclusions obtained in this study are reasonable.
 - To address your concern, we added analysis of future changes of T_w across China using ensemble mean of same CMIP6 experiments in the attribution analyses to verify the conclusion of this study. We found that spatial distribution of T_w across whole China are consistent with downscaling simulations (Fig. R1.25A, B, C). Changes of T_w under the SSP2-4.5 and SSP5-8.5 scenarios also displays similar spatial diversity that T_w in northern China increases faster than that in southern China (Fig. R1.25D, E).
 - To avoid confusion, we added more transition sentences to enhance the logic of the manuscript.
 - “Given the advantage of high resolution and better regional details, WRF-Chem downscaling model results were used for future projections of heat stress in China.”
 - “A five-year average could be influenced by internal climate variability, leading to uncertain projections. Using data from the CMIP6 experiments listed in Table S2, we reveal significant increasing trends in T_w across China under the SSP245 and SSP585 scenarios from 2015 to 2100 (Fig. S20). Notably, we observe a pronounced south-north difference, particularly under the SSP585 scenario, indicating that the non-uniform enhancement of T_w due to global warming is reasonably robust. Regional downscaling covering longer decades would contribute to better projections for China if computational resources permit.”

Fig. R1.24. Spatial distribution of geopotential height. Spatial distribution of summertime average geopotential height over 2010-2014 at 100 hPa (A), 500 hPa (C), 700 hPa (E) and 850 hPa (G) from FNL. Spatial distribution of summertime average geopotential height over 2010-2014 at 100 hPa (B), 500 hPa (D), 700 hPa (F) and 850 hPa (H) from ERA5.

Fig. R1.25. Spatial distribution of T_w and its future shifts. Spatial distribution of summertime average T_w over 2010-2014 (A) and over 2096-2100 under the SSP2-4.5 (B) and SSP5-8.5 (C) scenarios from CMIP6 ensemble means. Spatial distribution of differences in summertime average T_w between SSP2-4.5 (D), and Hist and SSP5-8.5 and Hist (E) from CMIP6 ensemble means.

Reference:

- Xu, Z., Han, Y., Tam, C.Y., Yang, Z.L. & Fu, C. Bias-corrected CMIP6 global dataset for dynamical downscaling of the historical and future climate (1979–2100). *Scientific Data* 8(1), 293 (2021).
- Wu, H., Lei, H., Lu, W. & Liu, Z. Future changes in precipitation over the upper Yangtze River basin based on bias correction spatial downscaling of models from CMIP6. *Environmental Research Communications* 4(4), 045002 (2022).
- Maraun, D. Bias correcting climate change simulations—a critical review. *Current Climate Change Reports* 2, 211-220 (2016).

Reviewer #2:

Comments:

Comments on the manuscript entitled “Widely-spread and uniformly elevated future heat stress in China driven by spatially heterogeneous response of water vapor” by Wang et al submitted to Nature Communications.

The authors investigated the changes in wet bulb temperature (T_w) using observed temperature and relative humidity data over the period 1979-2018. They have found that T_w has experienced a more significant increase in northern China compared to southern China, primarily due to spatially heterogeneous changes in water vapor pressure. Notably, such changes can lead to heightened thermal stress, thereby potentially impacting human health and adaptation measures in eastern China. This study is likely to generate immediate interest among professionals in climate- and health-related disciplines.

Nevertheless, there are some limitations to address. This study only used the observational data after 1979. The observational data can be extended back to 1960 in China. The absence of homogenization of the raw observational data could influence the findings. Furthermore, the dynamical downscaling simulations and mechanism analysis presented in the manuscript do not offer sufficient support for the stated conclusions.

Given these limitations, I recommend substantial revisions before considering this manuscript for publication in Nature Communications. A homogenized observational data, along with additional evidence from climate model simulations and mechanism analysis, is necessary to strengthen the study's validity and impact:

Reply:

- Thank you for your valuable comments and suggestions, which significantly improve the quality of this manuscript. We have read through your comments very carefully and have addressed them in the revised manuscript.

Major comments:

1. **Data:** The clarity of data sources and processing methods is crucial for the robustness of the study. The authors should address the following points: (a) Clarify whether the observational data and CMIP6 data used are "daily" or "monthly."

Reply:

- Sorry for the vagueness. We used daily observational data and monthly CMIP6 data in this study, and we have clarified them in Materials and Methods in the revised

- manuscript.
- “Surface observations of daily air temperature (T) and relative humidity (RH) at 2 meters from 1979 to 2018 were obtained from weather stations maintained by the China Meteorological Administration (CMA).”
 - “The historical simulations of monthly T and RH driven by time evolving all forcings and single forcing from the Detection and Attribution Model Intercomparison Project (DAMIP) and the Land Use Model Intercomparison Project (LUMIP) in the Coupled Model Intercomparison Project Phase 6 (CMIP6) were employed in this study to assess climate response to individual forcings, including GHG-only (GHG), aerosols-only (AER), natural-only (NAT) and land use-only (LU).”

(b) Please justify utilizing only observational data after 1979. Note that daily temperature and relative humidity data in China since 1960 have been accessible.

Reply:

- Thank you for your kind reminder of accessible data before 1979 in China. We used data after 1979 for two main reasons. Firstly, the number of surface weather stations in China greatly increased after 1979. As shown in Fig. R2.1A, the number of stations increased from below 780 to ~830. Secondly, the trend in T_w variations displayed a generally decreasing pattern prior to late-1970s but shifted to an increasing trend after late-1970s (Fig. R2.1B).
- According to our findings, this shift is largely attributed to the pronounced trend of global warming, as indicated by more noticeable surface temperature anomaly since late 1970s (Fig. R2.1C).
- Global surface temperature data shown in Fig. R2.1C were obtained from the National Centers for Environmental information (<https://www.ncei.noaa.gov/access/monitoring/climate-at-a-glance/global/time-series>).
- Following your comment, we added these reasons to avoid confusion in the revised manuscript.
- “We selected this study period because the number of surface weather stations increased after 1979 in obtained dataset (Fig. S23A), and there was a shift in the trend of T_w variations from a decreasing pattern to an increasing trend after the late-1970s (Fig. S23B), largely due to global warming indicated by surface temperature anomaly since late-1970s (Fig. S23C).”

Fig. R2.1. Site number, wet bulb temperature and surface temperature changes. (A) Number of surface observation stations. (B) Time series of average wet bulb temperature in China from 1960 to 2018. (C) Time series of surface temperature anomaly from 1850 to 2020.

(c) Temperature and relative humidity data should undergo homogenization, as many observational data contain breakpoints resulting from automated observation implementation and/or station relocation, etc. These may substantially affect the trend of temperature and relative humidity and lead to unreliable conclusions. The authors may consider using homogenized daily temperature and relative humidity data in China since 1960 to enhance data reliability (Argiriou, et al, 2023; Li et al, 2020).

Reply:

- Thank you for offering this great suggestion. Because homogenized air temperature provided by Argiriou et al. (2023) and relative humidity Li et al. (2020) contains very limited stations (around 300 compared to number of stations we used, Fig. R2.1A), particularly in Central and Western China (Fig. R2.2), which might not be able to reveal spatial patterns of T_w variations in China.
- To support revealed uniformly elevated trend of T_w across eastern China, we compared trend obtained from direct observations with the homogenized data you suggested. Although homogenized data shows higher T_w trend in southern China, different increasing rates of T_w between northern and southern China (0.2-0.3 K/decade higher in northern China) can still be found (Fig. R2.3).
- We further used two quality-controlled and homogenized datasets, the global station-based daily maximum wet-bulb temperature (GSDM-WBT) data from Dong et al. (2022) and the Met Office Hadley Centre Integrated Surface Database Humidity (HadISDH), to compare the T_w and E_a trends with direct observations in this study (Fig. R2.4C, D, G). They both show similar south-north difference in T_w and E_a trends with results in this study.
- In addition, we compared our observations with data obtained from the European Centre for Medium-Range Weather Forecasts (ECMWF) Reanalysis version 5 (ERA5) dataset (Fig. R2.4E, H) and the Modern-Era Retrospective analysis for Research and Applications, Version 2 (MERRA2) (Fig. R2.4F, I). They also exhibit similar spatial patterns of T_w and E_a trends, highlighting the representativeness of our findings obtained from direct observations.
- Following your suggestions, we have added these comparisons in the revised manuscript.
- “Homogenized daily surface relative humidity from Li et al. ⁷¹ and surface air temperature from Argiriou et al. ⁷² were employed to verify the trend of T_w and E_a derived from the original observations. We also adopted monthly T_w and E_a from the Met Office Hadley Centre Integrated Surface Database Humidity (HadISDH, <https://www.metoffice.gov.uk/hadobs/hadisdh>). HadISDH is a homogenized and quality controlled $5^\circ \times 5^\circ$ gridded global surface humidity dataset ⁷³. It is derived from a variety of sources, including weather station observations, satellite measurements, and reanalysis models. Daily maximum T_w of 1834 stations that had passed quality control and been homogenized were obtained from the global station-based daily maximum wet-bulb temperature (GSDM-WBT) data provided by Dong et al. ⁷⁴. Specific humidity and air temperature at 2 meters from the Modern-Era Retrospective analysis for Research and Applications, Version 2 (MERRA2) ⁷⁵ were adopted to calculate E_a and T_w . The spatial resolution of these variables was $0.625^\circ \times 0.5^\circ$.”
- “Surface homogenized humidity products and reanalysis datasets also exhibit similar spatial patterns of T_w trend (see Fig. S1, Fig. S2), confirming the capability of direct surface observations in depicting the major characteristics of spatial distributions of T_w variations.”

Fig. R2.2. Spatial distribution of surface observation sites. Spatial distribution of surface observation sites in this study (A) and homogenized dataset (B).

Fig. R2.3. Wet bulb temperature (T_w) variations. (A) Spatial distribution of T_w variations over 1979-2018 calculated using homogenized air temperature from Argiriou et al (2023) and relative humidity Li et al (2020). Only sites with significant trend ($P < 0.05$) are displayed. (B) Spatial distribution of T_w variations over 1979-2018 from observations in this study. Only sites with significant trend ($P < 0.05$) are displayed.

Fig. R2.4. Wet bulb temperature (T_w) and water vapor (E_a) variations. Spatial distribution of T_w (A) and E_a (B) variations over 1979-2018 from observations. Only sites with significant trend ($P < 0.05$) are displayed. (C) Spatial distribution of T_w variations over 1981-2018 from GSDM-WBT. Only sites with significant trend ($P < 0.05$) are displayed. Spatial distributions of T_w (D) and E_a (G) from HadISDH over 1979-2018. Black dots denote areas with significant trend ($P < 0.05$). Spatial distributions of T_w (E) and E_a (H) from ERA5 over 1979-2018. Black dots denote areas with significant trend ($P < 0.05$). Spatial distributions of T_w (F) and E_a (I) from MERRA2 over 1980-2018. Black dots denote areas with significant trend ($P < 0.05$).

Reference:

- Dong, J., Brönnimann, S., Hu, T., Liu, Y. and Peng, J., 2022. GSDM-WBT: global station-based daily maximum wet-bulb temperature data for 1981–2020. *Earth System Science Data*, 14(12), pp.5651-5664.
- Li, Z., Yan, Z., Zhu, Y., Freychet, N. and Tett, S., 2020. Homogenized daily relative humidity series in China during 1960–2017. *Advances in Atmospheric Sciences*, 37, pp.318-327.
- Argiriou, A.A., Li, Z., Armaos, V., Mamara, A., Shi, Y. and Yan, Z., 2023. Homogenised Monthly and Daily Temperature and Precipitation Time Series in China and Greece since 1960. *Advances in Atmospheric Sciences*, 40(7), pp.1326-1336.

2. Method: The methodology section requires additional clarification/additional numerical simulation and estimation of uncertainty in T_w projection. Address the following concerns:

(a) The authors conducted a set of 36km WRF-Chem simulations over the historical (2010-2014) and future (2096-2100) periods to investigate the future changes in T_w . However, the historical simulation was driven by FNL analysis data, while the future simulation was driven by bias-corrected CMIP6 data. Thus, the difference in WRF-Chem simulations between future and historical periods can be partly attributed to different sources of large-scale forcing data rather than climate change alone.

Reply:

- Thank you for raising this great question. Our WRF-Chem simulations were driven by the bias-corrected CMIP6 data, which underwent bias-correction using the ERA5 dataset (Xu et al., 2021). It took advantage of the non-linear trend of the ensemble mean of 18 CMIP6 models to give a more reliable projection of the long-term climate trend by subtracting mean bias of the long-term trend of the CMIP6 data relative to that of the ERA5 dataset over the historical time period. This can largely enhance the consistency of historical data and future data, making them comparable.
- Our present day WRF-Chem simulations were driven by FNL, while ERA5 and FNL are generally consistent due to constraints of surface observations, radiosonde data, satellite data, buoy data, and aircraft data. To address your concern, we compared them and found consistent features (Fig. R2.5).
- To further address your concern, we also added future projections using CMIP6 experiments directly. Spatial distribution of T_w changes across whole China from ensemble mean of same CMIP6 experiments in the attribution analyses are consistent with downscaling simulations (Fig. R2.6A, B, C). Changes of T_w under the SSP2-4.5 and SSP5-8.5 scenarios also display similar spatial diversity that T_w in northern China increases faster than that in southern China (Fig. R2.6D, E).
- All these additional analysis have been added in the revised manuscript.

Reference

- Xu, Z., Han, Y., Tam, C.Y., Yang, Z.L. and Fu, C., 2021. Bias-corrected CMIP6 global dataset for dynamical downscaling of the historical and future climate (1979–2100). *Scientific Data*, 8(1), p.293.

Fig. R2.5. Spatial distribution of geopotential height. Spatial distribution of summertime average geopotential height over 2010-2014 at 100 hPa (A), 500 hPa (C), 700 hPa (E) and 850 hPa (G) from FNL. Spatial distribution of summertime average geopotential height over 2010-2014 at 100 hPa (B), 500 hPa (D), 700 hPa (F) and 850 hPa (H) from ERA5.

Fig. R2.6. Spatial distribution of T_w and its future shifts. Spatial distribution of summertime average T_w over 2010-2014 (A) and over 2096-2100 under the SSP2-4.5 (B) and SSP5-8.5 (C) scenarios from CMIP6 ensemble means. Spatial distribution of differences in summertime average T_w between SSP2-4.5 (D), and Hist and SSP5-8.5 and Hist (E) from CMIP6 ensemble means.

(b) Specify if the projections in Figs. 5, 6, S9, and S10 are based on the WRF-Chem simulations or the raw CMIP6 datasets.

Reply:

- Fig. 5, 6, S9 showing surface T_w , T and E_a were based on downscaling results using WRF-Chem, while Fig. S10, S11 showing large-scale circulation patterns were based on raw CMIP6 datasets.
- We have specified them in the captions of these figures, following your suggestions.

(c) Examine whether CMIP6 and WRF-Chem simulations exhibit similar trends in T_w . The spread among individual CMIP6 simulations and WRF-Chem is essential for estimating the uncertainty of changes in T_w , E_a , and T . It is necessary to include an analysis of the uncertainty associated with the T_w projection using different sources of data.

Reply:

- Thank you for providing this valuable suggestion. In the revised manuscript, we added also trends in T_w from the CMIP6 experiments adopted in the attribution analysis. We found that spatial distribution of T_w across whole China are consistent with downscaling simulations (Fig. R2.7A, B, C). Although changes of T_w under the SSP2-4.5 and SSP5-8.5 scenarios are relatively smaller compared to WRF-Chem simulations, they also display similar spatial diversity that T_w in northern China increased faster than that in southern China (Fig. R2.7D, F).
- Following your suggestion of including analysis of the uncertainty, we found

- average changes of T_w from all 12 CMIP6 models are 2.29 ± 0.69 °C and 2.02 ± 0.49 °C under the SSP2-4.5 scenario; 4.64 ± 1.31 °C and 4.17 ± 1.25 °C under the SSP5-8.5 scenario in northern and southern China, respectively. Trends of T_w from the CMIP6 experiments can support the robustness of downscaling projections.
- CMIP6 models have limitations when applied to a region or a city due to the limited representation of spatial details on local scales (Zhao et al., 2022; Wu et al., 2022). Therefore, we adopted downscaling simulation to investigate future T_w projection in this study. Previous studies have confirmed that downscaling can provide more accurate surface meteorological variables, both spatially and temporally, compared with global climate models and earth system models (Seker and Gumus, 2022; Wei et al., 2022; Liang et al., 2006; Chapman et al., 2023; Gebrechorkos et al., 2023).
 - We have added this analysis in the revised manuscript.
 - “Spatial distribution of T_w trends across whole China from the CMIP6 experiments adopted in the attribution analysis are consistent with downscaling simulations (Fig. S17A, B, C). Average changes of T_w from all 12 CMIP6 models are 2.29 ± 0.69 °C and 2.02 ± 0.49 °C under the SSP2-4.5 scenario (Fig. S17D, E); 4.64 ± 1.31 °C and 4.17 ± 1.25 °C under the SSP5-8.5 scenario (Fig. S17F, G) in northern and southern China, respectively. Although changes of T_w under the SSP2-4.5 and SSP5-8.5 scenarios from the CMIP6 are relatively smaller compared to WRF-Chem simulations due to its limited representation of spatial details on local scales^{48,49}, they display similar spatial diversity that T_w in northern China increases faster than that in southern China.”

Reference:

- Chapman, S., Syktus, J., Trancoso, R., Thatcher, M., Toombs, N., Wong, K.K.H. and Takbash, A., 2023. Evaluation of Dynamically Downscaled CMIP6-CCAM Models Over Australia. *Earth's Future*, 11(11), p.e2023EF003548.
- Gebrechorkos, S., Leyland, J., Slater, L., Wortmann, M., Ashworth, P.J., Bennett, G.L., Boothroyd, R., Cloke, H., Delorme, P., Griffith, H. and Hardy, R., 2023. A high-resolution daily global dataset of statistically downscaled CMIP6 models for climate impact analyses. *Scientific Data*, 10(1), p.611.
- Liang, X.Z., Pan, J., Zhu, J., Kunkel, K.E., Wang, J.X. and Dai, A., 2006. Regional climate model downscaling of the US summer climate and future change. *Journal of Geophysical Research: Atmospheres*, 111(D10).
- Seker, M. and Gumus, V., 2022. Projection of temperature and precipitation in the Mediterranean region through multi-model ensemble from CMIP6. *Atmospheric Research*, 280, p.106440.
- Wei, X., Wang, G., Feng, D., Duan, Z., Hagan, D.F.T., Tao, L., Miao, L., Su, B. and Jiang, T., 2023. Deep-learning-based harmonization and super-resolution of near-surface air temperature from CMIP6 models (1850–2100). *International Journal of Climatology*, 43(3), pp.1461-1479.
- Wu, H., Lei, H., Lu, W. and Liu, Z., 2022. Future changes in precipitation over the upper Yangtze River basin based on bias correction spatial downscaling of models from CMIP6. *Environmental Research Communications*, 4(4), p.045002.
- Zhao, N., Jiao, Y. and Zhang, L., 2022. Projections of precipitation change from CMIP6 based on a new downscaling method in the Poyang Lake basin, China. *Journal of Hydrology: Regional Studies*, 42, p.101138.

Fig. 5. Spatial distribution of T_w and its future shifts. Spatial distribution of summertime average T_w over 2010-2014 (A) and over 2096-2100 under the SSP2-4.5 (B) and SSP5-8.5 (C) scenarios from WRF-Chem simulations. The red line indicates the location of the Heihe–Tengchong Line (and internationally as the Hu line) that divides the area of China into two parts with contrasting population densities. Spatial distribution of differences in summertime average T_w between SSP2-4.5 (D), and Hist and SSP5-8.5 and Hist (E) from WRF-Chem simulations.

Fig. R2.7. Spatial distribution of T_w and its future shifts. Spatial distribution of summertime average T_w over 2010-2014 (A) and over 2096-2100 under the SSP2-4.5 (B) and SSP5-8.5 (C) scenarios from CMIP6 ensemble means. (D) Spatial distribution of differences in summertime average T_w between SSP2-4.5 and Hist from CMIP6 ensemble means. (E) Average changes of T_w in China, northern China (NC) and southern China (SC) between SSP2-4.5 and Hist from ensemble mean of the 12 CMIP6 models listed in Table S2. The error bars show one standard deviation of the multimodal ensemble. (F) Spatial distribution of differences in summertime average T_w between SSP5-8.5 and Hist from CMIP6 ensemble means. (G) Average changes of T_w in China, northern China (NC) and southern China (SC) between SSP5-8.5 and Hist from ensemble mean of the 12 CMIP6 models listed in Table S2. The error bars show one standard deviation of the multi-model ensemble.

3. Mechanism: The authors must provide a more comprehensive understanding of the mechanisms driving historical and future changes in water vapor pressure (E_a) and wet bulb temperature (T_w). Consider addressing the following concerns:

(a) The authors attributed the historical change in water vapor pressure (E_a) and wet bulb temperature (T_w) to the changes in the South Asian High (SAH) and Western Pacific Subtropical High (WPSH). However, in addition to the SAH and WPSH, other factors, e.g. the Asian summer monsoon, ENSO, and PDO, also play important roles in modulating the East Asian climate. Please elaborate on the extent to which the SAH and WPSH influence E_a and T_w compared to other factors.

Reply:

- The SAH and WPSH are atmospheric circulation systems that can directly exert influences on regional weather and moisture transport. They are direct manifestations of large-scale atmospheric circulations influenced by the Asian summer monsoon, ENSO, and PDO, which in turn modulate regional weather and climate (Liu et al., 2019; Dong, 2016; Xu et al., 2022). That is to say, the impacts of the Asian summer monsoon, ENSO, and PDO are embedded in the patterns of SAH and WPSH incorporate.
- To address your concern, we investigated the relationship between ENSO, PDO and the Asian summer monsoon, and variations of T_w and E_a . Time series analysis of the Oceanic Nino Index (ONI) and the East Asia Summer Monsoon Index (EASMI) from 1979 to 2018 reveal no significant trend (Fig. R2.8A, B). However, the PDO Index (PDOI) exhibits a general decreasing trend, with a positive phase observed before 2000 and a negative phase after 2000 (Fig. R2.8C). The ONI and PDOI were obtained from the National Centers for Environment Information (NCEI, <https://www.ncei.noaa.gov/access/monitoring/products/>). The EASMI was obtained from Li and Zeng (2002, 2003). (<http://lijianping.cn/dct/page/65577>).
- We further calculated correlation coefficients between the ONI, PDOI, EASMI, and variations of T_w and E_a . The ONI shows a weak correlation with T_w and E_a in the Central region of China, with absolute correlation coefficients mostly below 0.4 (Fig. R2.9A, D). The PDOI exhibits a strong correlation with variations of T_w and E_a at most sites across China, with absolute correlation coefficients exceeding 0.4 ($P < 0.05$) (Fig. R2.9C, F). Specifically, the PDOI displays a positive correlation with T_w and E_a in southern regions, while a negative correlation is observed in northern regions. The negative phase of PDO has been linked with northward shift of the WPSH (Tong et al., 2021; Ye et al., 2015; Matsumura and Horinouchi, 2016), partially contributing to faster increase in T_w in northern China.
- To further address your concern and following your suggestion of elaborating the impacts, we conducted an empirical orthogonal function (EOF) analysis on T_w during the period of 1979-2018 to find the underlying mechanism and factors causing T_w variations in preceding decades (Fig. R2.13). The first mode of EOF, accounting for 42.7% of the total variances, shows consistent T_w changes across China with observed trend of T_w variations, characterized by faster increasing rate in northern China. In line with the significant positive changes revealed by

- regression of T_s on PC1 (Fig. R2.11A), T_s also shows significantly increasing trend during the period of 1979-2018 (Fig. R2.11B) over the high-latitude land region of East Asia. The rapid warming in this arid region may be due to its low heat capacity (Ji et al., 2014) and lower anthropogenic aerosol emissions compared to lower latitudes (Samset et al., 2018). These observations can be further supported by the findings of surface air temperature changes under GHG-only and aerosols-only forcing outputs from the CMIP6 experiments (Fig. R2.12).
- It implies that the intensified surface temperature over this region positively contributes to the dominant mode of T_w variations. During the study period, mean surface temperature in summer within the high-latitude land region of East Asia displays a notably increasing trend of $0.51\text{ }^\circ\text{C decade}^{-1}$ (Fig. R2.11C), which is significantly correlated with PC1 (0.73, $p < 0.01$) (Fig. R2.11D). Faster warming of the high-latitude region results in upper atmospheric pressure heightened, which leads to anticyclone enhancement in northern regions at upper levels (Fig. R2.13A, B). Anticyclone enhancement in northern regions triggers the eastward and northward extension of the SAH and westward and northward marching of the WPSH, which are typically situated over the Tibetan region and the Western Pacific Ocean in summer, respectively (Fig. R2.13A, B). Increased influences of the SAH and WPSH over larger swathes of southern China lead to descending motion prevailing over southern China at lower levels, suppressing moisture transport from both the Bay of Bengal and the South China Sea but accelerating moisture transport from the Pacific Ocean to northern China (Fig. R2.13C, D). Correlation and composite analyses highlight the strong connection between T_w , and the SAH and WPSH as T_w tends to be higher in northern regions but lower in southern regions when the SAH and WPSH are in higher latitudes (Fig. R2.14, Fig. R2.15).
 - We have added these analyses in the revised manuscript.
 - “To understand the underlying mechanism and factors causing T_w variations in preceding decades, we conducted an empirical orthogonal function (EOF) analysis on T_w during the period of 1979-2018. The first leading mode of EOF, accounting for 42.7% of the total variance, shows consistent T_w changes across China with observed trend of T_w variations, characterized by faster increasing rate in northern China (Fig. S6A). In line with the significant positive changes revealed by regression of T_s on PC1 (Fig. 4A), T_s also shows significantly increasing trend during the period of 1979-2018 (Fig. 4B) over the high-latitude regions of East Asia. The more rapid warming in this arid region may be due to its low heat capacity 23 and lower anthropogenic aerosol emissions compared to lower latitudes 46. These observations can be further supported by the findings of surface air temperature changes under GHG-only and aerosols-only forcing outputs from the Coupled Model Intercomparison Project Phase 6 (CMIP6) experiments (Fig. S7). It implies that the intensified surface temperature over this region positively contributes to the dominant mode of T_w variations. During the study period, mean surface temperature in summer within the high-latitude 1 regions of East Asia displays a notably increasing trend of $0.51\text{ }^\circ\text{C decade}^{-1}$ (Fig. 4C), which is significantly correlated with PC1 (0.73, $p < 0.01$) (Fig. 4D). Faster warming of the high-latitude region results in upper atmospheric pressure heightened, which leads to anticyclone enhancement in northern regions at upper levels (Fig. S8A, B). Anticyclone

enhancement in northern regions triggers the eastward and northward extension of the South Asia high (SAH, represented by 16760-dagpm line on 100 hPa 47) (Fig. S9A, B), and westward and northward marching of the western Pacific subtropical high (WPSH, represented by 5880-dagpm line on 500 hPa 48) (Fig. S9C, D), which are typically situated over the Tibetan region and the Western Pacific Ocean in summer, respectively (Fig. S8A, B). Increased influences of the SAH and WPSH over larger swathes of southern China lead to descending motion prevailing over southern China at lower levels, suppressing moisture transport from both the Bay of Bengal and the South China Sea but accelerating moisture transport from the Pacific Ocean to northern China (Fig. S8C, D). Correlation and composite analyses further verify the strong connection between Tw, and the SAH and WPSH as Tw tends to be higher in northern regions but lower in southern regions when the SAH and WPSH are in higher latitudes (Fig. S10, Fig. S11). This results in declined water vapor in the atmosphere, and further suppresses precipitation, leading to less water content of the ground that can be evaporated (Fig. 3D). Historical changes of geopotential height and atmospheric circulation are highly consistent with regression results (Fig. S12, Fig. S13), highlighting their contributions to Tw variations.”

Fig. R2.8. ENSO, Asia summer monsoon and the Pacific Decadal Oscillation (PDO) changes. Time series of the Oceanic Niño Index (ONI) (A), the East Asia Summer Monsoon Index (EASMI) (B), and the PDO Index (PDOI) (C) during the period from 1979 to 2018.

Fig. R2.9. Correlations of the ONI, EASMI and PDOI with variations of T_w and E_a . Correlations of the ONI (A), EASMI (B), and PDOI (C) with variations of T_w . Correlations of the ONI (D), EASMI (E), and PDOI (F) with variations of E_a . Only sites with significant trend ($P < 0.05$) are displayed.

Fig. R2.10. Empirical orthogonal function (EOF) analysis of T_w . Spatial (A) and temporal variations (B) of the first leading mode inferred by EOF analysis.

Fig. R2.11. Connection between surface temperature (T_s) and the first leading mode (PC1) of empirical orthogonal function. (A) Regression of T_s on the first leading mode. Black dots denote areas with significant correlation ($P < 0.05$). (B) T_s trend with variations of T_w . Correlations of the longitudes of the SAH eastward ridge point (ERP) (C) Time series of PC1 and average T_s anomaly of the northern region in East Asia (33°N-50°N, 80°E-120°E, red boxes in A and B). The scatter plot between T_s anomalies in NA and PC1 in each year from 1979 to 2018.

Fig. R2.12. Global surface temperature variations. Spatial distribution of surface air temperature variations during the period from 1979 to 2014 under GHG-only (A) and aerosols-only (B) forcing conditions. Black dots denote areas with significant trend ($P < 0.05$).

Fig. R2.13. Regression of atmospheric features on the first leading mode. Regression of geopotential height and circulation at 100 hPa (A), 500 hPa (B), 700 hPa (C) and 850 hPa (D) on the first leading mode. Red dots denote areas with significant correlation ($P < 0.05$). Blues solid lines in A and B indicate climatologically averaged locations of the SAH (represented by 16760-dagpm line) and the WPSH (represented by 5880-dagpm line), respectively. Blues dashed lines in A and B indicate varied locations of the SAH and WPSH, respectively.

Fig. R2.14. Correlations of the SAH and WPSH with variations of T_w . Correlations of the latitudes of the SAH eastward ridge point (ERP) (A) and WPSH westward ridge point (WRP) (B) with variations of T_w . Correlations of the longitudes of the SAH eastward ridge point (ERP) (C) and WPSH westward ridge point (WRP) (D) with variations of T_w . Only sites with significant trend ($P < 0.05$) are displayed.

Fig. R2.15. Composite analysis of T_w . The composite differences of T_w between northward and southward movement of the SAH ERP (A) and WPSH WRP (B). The composite differences of T_w between westward and eastward movement of the SAH ERP (C) and WPSH WRP (D).

Reference:

- Dong, X. Influences of the Pacific decadal oscillation on the east Asian summer monsoon in non - ENSO years. *Atmospheric Science Letters* 17(1), 115-120 (2016).
- Ji, F., Wu, Z., Huang, J. and Chassignet, E.P., 2014. Evolution of land surface air temperature trend. *Nature Climate Change*, 4(6), pp.462-466.
- Jianping, L. & Qingcun, Z. A new monsoon index and the geographical distribution of the global monsoons. *Advances in atmospheric sciences* 20, 299-302 (2003).
- Li, J. & Zeng, Q. A unified monsoon index. *Geophysical Research Letters* 29(8), 1274 (2002).
- Liu, Q., Zhou, T., Mao, H. & Fu, C. Decadal variations in the relationship between the western Pacific subtropical high and summer heat waves in East China. *Journal of climate* 32(5), 1627-1640 (2019).
- Matsumura, S. & Horinouchi, T. Pacific Ocean decadal forcing of long-term changes in the western Pacific subtropical high. *Scientific reports* 6(1), 37765 (2016).
- Samset, B.H., Sand, M., Smith, C.J., Bauer, S.E., Forster, P.M., Fuglestedt, J.S., Osprey, S. and Schleussner, C.F., 2018. Climate impacts from a removal of anthropogenic aerosol emissions. *Geophysical Research Letters*, 45(2), pp.1020-1029.
- Tong Q., Huang Y., Duan M. & Zhao Q. Possible contribution of the PDO to the eastward retreat of the western Pacific subtropical high. *Atmospheric and Oceanic Science Letters* 14,

- 100005 (2020).
- Xu, M., Xu, H., Ma, J. & Deng, J. Impact of Pacific Decadal Oscillation on interannual relationship between El Niño and South China Sea summer monsoon onset. *International Journal of Climatology* 42(5), 2739-2753 (2022).
 - Ye, T., Shen, Q., Wang, K., Zhang, Z. & Zhao, J. Interdecadal change of the northward jump time of the western Pacific subtropical high in association with the Pacific decadal oscillation. *Journal of Meteorological Research* 29(1), 59-71 (2015).
 - Yuan, Y., Gao, H. and Ding, T., 2020. The extremely north position of the western Pacific subtropical high in summer of 2018: Important role of the convective activities in the western Pacific. *International Journal of Climatology*, 40(3), pp.1361-1374.

(b) The provided explanation regarding the influence of the SAH and WPSH on E_a and T_w lacks persuasiveness. It is imperative to conduct additional investigations to elucidate the role of external forcing and internal climate variability in modulating changes in T_w . Please refer to the detailed comments provided in the following section for further insights.

Reply:

- Thanks for pointing out the inadequacy of our explanation. We acknowledge your point that the existing explanation lacks persuasiveness in fully validating the influence of the SAH and WPSH on E_a and T_w variations. We reorganized our analysis and associated the heterogeneity of E_a and T_w variations with faster warming of high-latitude regions of East Asia under global warming, which regulates large-scale atmospheric features and leads to extended impacts of the SAH and WPSH over southern China and suppressed moisture transport.
- We first conducted an empirical orthogonal function (EOF) analysis on T_w during the period of 1979-2018 (Fig. R2.13). The first mode of EOF, accounting for 42.7% of the total variances, shows consistent T_w changes across China with observed trend of T_w variations, characterized by faster increasing rate in northern China. In line with the significant positive changes revealed by regression of T_s on PC1 (Fig. R2.11A), T_s also shows significantly increasing trend during the period of 1979-2018 (Fig. R2.11B) over the high-latitude land region of East Asia. The rapid warming in this arid region may be due to its low heat capacity (Ji et al., 2014) and lower anthropogenic aerosol emissions compared to lower latitudes (Samset et al., 2018). These observations can be further supported by the findings of surface air temperature changes under GHG-only and aerosols-only forcing outputs from the CMIP6 experiments (Fig. R2.12).
- It implies that the intensified surface temperature over this region positively contributes to the dominant mode of T_w variations. During the study period, mean surface temperature in summer within the high-latitude land region of East Asia displays a notably increasing trend of $0.51\text{ }^\circ\text{C decade}^{-1}$ (Fig. R2.11C), which is significantly correlated with PC1 (0.73, $p < 0.01$) (Fig. R2.11D). Faster warming of the high-latitude region results in upper atmospheric pressure heightened, which leads to anticyclone enhancement in northern regions at upper levels (Fig. R2.13A, B). Anticyclone enhancement in northern regions triggers the eastward and northward extension of the SAH and westward and northward marching of the

WPSH, which are typically situated over the Tibetan region and the Western Pacific Ocean in summer, respectively (Fig. R2.13A, B). Increased influences of the SAH and WPSH over larger swathes of southern China lead to descending motion prevailing over southern China at lower levels, suppressing moisture transport from both the Bay of Bengal and the South China Sea but accelerating moisture transport from the Pacific Ocean to northern China (Fig. R2.13C, D). Correlation and composite analyses highlight the strong connection between T_w , and the SAH and WPSH as E_a tends to be higher in northern regions but lower in southern regions when the SAH and WPSH are in higher latitudes (Fig. R2.14, Fig. R2.15). We have added these analyses in the revised manuscript.

- The role of external forcing and internal climate variability in modulating changes in T_w have been discussed in attribution analysis in the original manuscript. We found that the external forcings of greenhouse gases and aerosols significantly contribute to variations of T_w (Fig. R2.16C, D). Land use changes also play a relatively minor role on T_w variations (Fig. R2.16F). The internal climate variability has relatively small impact on T_w variations (Fig. R2.16E). Following your suggestion for conducting additional investigations to elucidate the role of external forcing and internal climate variability in modulating changes in T_w , we additionally employed empirical decomposition method (EMD) to separate internal natural variability and external anthropogenic forcing in observations. The residual obtained from the EMD can be considered as the impacts of anthropogenic emissions, urbanization and land use change (Loehle and Scafetta, 2011). We found the natural variability exhibited insignificant trends in T_w at most sites in China (Fig. R2.17A). However, external forcings emerge as the dominant factor driving T_w changes during the period of 1979-2018 (Fig. R2.17B). This is evident from the consistent spatial patterns observed, which align with our original findings. We further employed empirical orthogonal function (EOF) decomposition on derived T_w values resulting from external forcings to identify the contributions of each anthropogenic factor including GHG, AER and LU (Fig. R2.18). EOF1 and EOF2 can be considered as the contributions of GHG and AER, respectively, as they exhibit similar spatial patterns as our attribution analysis (Fig. R2.17A, B, Fig. R2.16C, D). They account for 68.6% and 23.8% of the total variations, respectively, which closely aligns with the results obtained from our attribution analysis (GHG: ~63%, AER: ~24%). The third mode of EOF shows a generally decreasing trend of T_w , which can be attributed to the impacts of urbanization and land use changes (Fig. R2.18C).

Fig. R2.16. Observed and simulated variations of T_w . (A) Spatial distribution of observed T_w variations during the period of 1979-2014. Only sites with significant trend ($P < 0.05$) are displayed. (B) Spatial distribution of simulated T_w variations under all-forcing conditions during the period of 1979-2014. Black dots denote areas with significant trend ($P < 0.05$). Spatial distribution of simulated T_w variations under GHG-only (C), aerosols-only (D), natural-only (E) and land use-only (F) forcing conditions during the period of 1979-2014. Black dots denote areas with significant trend ($P < 0.05$).

Fig. R2.17. Empirical decomposition analysis on T_w . Spatial distribution of T_w trends caused by natural variabilities (A) and external forcings (B) during the period of 1979-2018.

Fig. R2.18. Empirical orthogonal function (EOF) decomposition of T_w . Spatial patterns of T_w of (A) EOF1, (B) EOF2, and (C) EOF3.

Other comments:

L55: replace “country” with “countries”

Reply:

➤ Revised as suggested.

L86-89: The criteria for categorizing stations into northern and southern stations using 33N are unclear. Considering the observed tri-pole pattern of summertime precipitation over eastern China in the second half of the 20th century, particularly in the North China-Yangtze River Basin-South China regions (e.g. Ding et al., 2008), further details on the rationale for station classification are important to enhance the robustness of the analysis. To my knowledge, the precipitation trend shows a dry (North China)-wet (Yangtze River Basin)-dry (South China) pattern over eastern China in summer over the

past 60 years or so. Why does the water vapor pressure (E_a) increase in North China and decrease in South China (Fig. 2c)?

Reply:

- Thank you for the professional question. The categorization of stations into northern and southern regions was employed to analyze the disparities in T_w and E_a variations between these two areas. In China, the Qinling-Huaihe line serves as the north-south dividing line, spanning from approximately 32° to 34°N. Given the apparent differences in T_w trends between the north and south, the selection of 33°N as the dividing line in this study is considered reasonable.
- Precipitation patterns do not directly reveal surface water vapor changes as surface water vapor is mainly influenced by evaporation and moisture transport (Sherwood et al., 2010; Yan et al., 2020; Zhang and Datta, 2007). In our study period, significant increasing trends in evaporation are identified in northern China (Fig. R2.19A). However, southern China witnesses a decreasing trend in evaporation (Fig. R2.19A) alongside a concurrent trend of moisture divergence (Fig. R2.19B). These two factors synergistically lead to a reduction in E_a within that region.

Reference:

- Sherwood, S.C., Roca, R., Weckwerth, T.M. and Andronova, N.G., 2010. Tropospheric water vapor, convection, and climate. *Reviews of Geophysics*, 48(2).
- Yan, H., Huang, J., He, Y., Liu, Y., Wang, T. and Li, J., 2020. Atmospheric water vapor budget and its long-term trend over the Tibetan Plateau. *Journal of Geophysical Research: Atmospheres*, 125(23), p.e2020JD033297.
- Zhang, J. and Datta, A.K., 2004. Some considerations in modeling of moisture transport in heating of hygroscopic materials. *Drying Technology*, 22(8), pp.1983-2008.

Fig. R2.19. Change in meteorological variables. Spatial distributions of changes in surface evaporation (A), vertical integrated moisture divergence (B), surface temperature (C) and total precipitation (D) over 1979-2018. Black dots denote areas with significant trend ($P < 0.05$).

L99-100: The authors concluded that E_a plays a more important role than T in determining the increase in T_w . However, the means of establishing this relative importance remain unclear. Comparison of Fig. 1B, Fig. 2A, and Fig. 2C suggests that T , conversely, appears to play a more pivotal role. Both T and T_w show a widespread increase in China with a pronounced trend in northern China and a less pronounced trend in the south.

Reply:

- Sorry for the vagueness. We investigated changes of T_w when changed T or E_a only but keep the value of the other factor at year 1979 (Fig. R2.20). When E_a changes with time but T is fixed at year 1979, T_w shows more similar variations with observed patterns in southern China, displaying negative trends (Fig. R1.1B, Fig. R2.20D). This suggests that the different trends observed in Southern China is more likely to be caused by E_a changes (Fig. R2.20A). While in northern China, T dominates still changes of T_w .

- To avoid confusion, we changed all related expressions in the revised manuscript.
- “Changes in T_w when either T or E_a varied but the other factor fixed at year 1979 highlight that E_a changes are responsible for the observed different T_w trends in southern China, while T dominates still changes of T_w in northern China (Fig. S4), but T dominates still changes of T_w in northern China.”

Fig. R2.20. Air temperature (T), water vapor (E_a) and induced T_w variations. (A) Spatial distribution of T variations during the period from 1979 to 2018. Only sites with significant trend ($P < 0.05$) are displayed. (B) Spatial distribution of E_a variations during the period from 1979 to 2018. Only sites with significant trend ($P < 0.05$) are displayed. (C) Spatial distribution of T induced T_w variations during the period from 1979 to 2018. Only sites with significant trend ($P < 0.05$) are displayed. (D) Spatial distribution of E_a induced T_w variations during the period from 1979 to 2018. Only sites with significant trend ($P < 0.05$) are displayed.

L112-113: The summertime climate in East Asia is influenced not only by the South Asia High (SAH) and the Western Pacific subtropical High (WPSH) but also by factors such as the East Asian summer monsoon, ENSO, and PDO.

Reply:

- This comment is associated with the one mentioned in 3(a). Please refer to the response above.

L118-127: I'm not convinced by the author's rationale for the decline in water vapor over southern China linked to changes in SAH and WPSH. Notably, the most substantial increases in geopotential height occur beyond the core regions of SAH and WPSH, around 45N and 60N. The eastward movement of SAH and the westward shift of WPSH, as depicted in Fig. S3, are not clearly discernible. Global warming induces an overall elevation in geopotential height; hence, a marginal increase in East Asia cannot be equated with an eastward or westward shift of high-pressure systems. Furthermore, atmospheric circulation is closely related to the gradient of geopotential height (GH), not GH itself. The overall GH increase may not affect the GH gradient, circulation, or water vapor divergence. Additionally, given that most atmospheric moisture resides below 700hPa, the focus on mid and upper-tropospheric changes rather than the lower troposphere lacks justification. Lastly, it is crucial to explore if the eastward ridge point of SAH and the westward ridge point of WPSH in Fig. 4 significantly correlate with T_w changes.

Reply:

- Thank you for raising your concern about our rationale for the decline in water vapor over southern China linked to changes in SAH and WPSH. To address your concern and following your suggestions, we reorganized our analysis and revealed the mechanism that external forcing accelerates warming in high-latitude regions, which reshapes large-scale circulations over East Asia. As a result, the influence of the SAH and WPSH is extended, ultimately leading to changes in E_a in northern and southern China. Please refer to our response to 3(b) above for more detailed explanations.
- Global warming induces an overall elevation in geopotential height (GH), but the most substantial increases in GH occur beyond the core regions of SAH and WPSH, leading to the changes of the gradients of GH over East Asia (Fig. R2.21, Fig. R2.22A, B). Therefore, we observed an eastward extension of SAH in summer (Fig. R2.23A). and a notable westward extension (Fig. R2.23C) coupled with a slight northward shift (Fig. R2.23D) of WPSH. The movements of the SAH and WPSH due to global warming in recent decades have also been documented in other studies (Huang et al., 2015; Zhou et al., 2009; Ren et al., 2015; Zhang et al., 2017). This results in changes of atmospheric circulations over this region thus regulating moisture transport. Given that most atmospheric moisture resides below 700hPa, we analyzed atmospheric circulation changes at 700 hPa (Fig. R2.22C) and 850 hPa (Fig. R2.22D) and found a decrease of moisture transport from both the India Ocean and the Pacific Ocean to southern China but an increase from the Pacific Ocean to northern China, inducing a faster enhancement of T_w in northern China than that in southern China.
- We have added a more comprehensive explanation in the revised manuscript.

Fig. R2.21. Atmospheric systems and their evolutions. Spatial distribution of average summertime geopotential height on 100 hPa (A), 500 hPa (C), 700 hPa (E) and 850 hPa (G) over the period from 1979 to 2018. Spatial distribution of geopotential height variations on 100 hPa (B), 500 hPa (D), 700 hPa (F) and 850 hPa (H) over the period from 1979 to 2018. Black dots denote areas with significant trend ($P < 0.05$).

Fig. R2.22. Changes of atmospheric circulations. Spatial distribution of differences in average summertime geopotential height and atmospheric circulation on 100 hPa (A), 500 hPa (B), 700 hPa (C) and 850 hPa (D) between the period of 1979-1998 and 1999-2018.

Fig. R2.23. Variations of locations of SAH and WPSH systems. Time series of monthly longitude (A) and latitude (B) of the eastward ridge point (ERP) of the South Asia high

(SAH) over the period from 1979 to 2018. Time series of monthly longitude (C) and latitude (D) of the westward ridge point (WRP) of the western Pacific subtropical high (WPSH) over the period from 1979 to 2018.

Reference:

- Ren, X., Yang, D. and Yang, X.Q., 2015. Characteristics and mechanisms of the subseasonal eastward extension of the South Asian high. *Journal of Climate*, 28(17), pp.6799-6822.
- Huang, Y., Wang, H., Fan, K. and Gao, Y., 2015. The western Pacific subtropical high after the 1970s: westward or eastward shift?. *Climate Dynamics*, 44, pp.2035-2047.
- Zhou, T., Yu, R., Zhang, J., Drange, H., Cassou, C., Deser, C., Hodson, D.L., Sanchez-Gomez, E., Li, J., Keenlyside, N. and Xin, X., 2009. Why the western Pacific subtropical high has extended westward since the late 1970s. *Journal of Climate*, 22(8), pp.2199-2215.
- Zhang, J., Tang, Q., Chen, H. and Liu, S., 2017. Northward shift in circulation system over the Asian mid - latitudes linked to an increasing heating anomaly over the northern Tibetan Plateau during the past two decades. *International Journal of Climatology*, 37(2), pp.834-848.

L143: replace “predicted T_w ” with “projected T_w ”. Please note the difference between “prediction” and “projection”.

Reply:

- Done as suggested.

L155: The expression “divergence in E_a ” is confusing, a rephrasing is necessary for clarity.

Reply:

- We rephrased it to “there are notable spatial difference in E_a changes”.

L157-158: Again, an increase in pressure does not inherently influence circulation; it is the pressure gradient that holds significance. The authors attempt to elucidate the change in E_a (2-m water vapor pressure) using Figs. S10 and S11, where 100hPa and 500hPa pressure/circulation are showcased. It is advisable to investigate lower tropospheric circulation and moisture transfer, as an overall increase in GH might not impact the GH gradient, circulation, or water vapor divergence.

Reply:

- Thank you for offering this great suggestion. We investigated lower tropospheric circulations at 700 hPa and 850 hPa and found anticyclone enhancements over the Bay of Bangladesh and South China Sea at lower layers (Fig. R2.24, Fig. R2.25G, H). Influenced by the air flow at the ridge of anticyclone, more water vapor transports to the Pacific Ocean instead of to southern China. Moisture transported from the Pacific Ocean to northern China was enhanced at middle and lower layers (Fig. R2.25C, D, G, H). We have added these analyses in the revised manuscript.

- “Under a warming climate, both SAH and WPSH exhibit substantial intensification, with a notable increase in central pressure (Fig. S17), resulting in anticyclone enhancements over the Bay of Bangladesh and South China Sea at lower layers (Fig. S19E, F, G, H). As a result, influenced by the air flow at the ridge of anticyclone, more water vapor is transported to the Pacific Ocean instead of to southern China, different from the enhanced moisture transport from the Pacific Ocean to northern China (Fig. S19).”

Fig. R2.24. Historical and future changes of atmospheric circulations. Spatial distribution of average summertime geopotential height and circulation on 700 hPa over the Hist period (2010-2014) (A) and future period (1996-2100) under the SSP2-4.5 (C) and SSP5-8.5 (E) scenarios from bias corrected CMIP6 global dataset. Spatial distribution of average summertime geopotential height on 850 hPa over the Hist period (2010-2014) (B) and future period (1996-2100) under the SSP2-4.5 (D) and SSP5-8.5 (F) scenarios from bias corrected CMIP6 global dataset.

Fig. R2.25. Future changes of atmospheric circulations. Spatial distribution of future changes of summertime geopotential height and circulation at 100 hPa under the SSP2-4.5 (A) and SSP5-8.5 (B) scenarios from bias corrected CMIP6 global dataset. Spatial distribution of future changes of summertime geopotential height and circulation at 500 hPa under the SSP2-4.5 (C) and SSP5-8.5 (D) scenarios from bias corrected CMIP6 global dataset. Spatial distribution of future changes of summertime geopotential height and circulation at 700 hPa under the SSP2-4.5 (E) and SSP5-8.5 (F) scenarios from bias

corrected CMIP6 global dataset. Spatial distribution of future changes of summertime geopotential height and circulation at 850 hPa under the SSP2-4.5 (G) and SSP5-8.5 (H) scenarios from bias corrected CMIP6 global dataset.

Figs. 1-4: Clarify the significance test employed. Is the serial correlation considered in the significance test?

Reply:

- We employed Student T-test to test the significance of the serial data, and the serial correlation was considered in the significance test. We have added the clarification in Materials and Methods.
- “Student T-test was used to detect the statistical significance of the regression coefficient and correlation coefficient.”

Fig. 5D, 5E, Fig.S9: These figures illustrated the difference in T , E_a , T_w between 2096-2100 and 2010-2014. A five-year average might be strongly influenced by internal climate variability, amplifying projection uncertainty. Recommending the use of a 30-year average or a long-term trend for more robust projections.

Reply:

- Thanks for your valuable recommendation. Using a 30-year average or a long-term trend requires at least 60-year downscaling simulation, which is far beyond our current computational capability given the much larger recourses needed for coupled regional aerosol-climate simulations.
- We agree that internal climate variability and inter-annual variability could make a difference. That’s why we tried to include as many years as we could in downscaling. To further address your concern, we investigated future variations in T_w using data from the same CMIP6 experiments as those adopted in attribution analysis.
- We found significant increasing trends in T_w across China under the SSP2-4.5 and SSP5-8.5 scenarios (Fig. R2.26). Notably, we observed a remarkable south-north difference, which was even more pronounced under the SSP5-8.5 scenario, suggesting that the non-uniform enhancement of T_w due to global warming is reliable.
- We acknowledged this as a limitation of our study and discussed it in Discussion in the revised manuscript.
- “A five-year average could be influenced by internal climate variability, leading to uncertain projections. Using data from the CMIP6 experiments listed in Table S2, we reveal significant increasing trends in T_w across China under the SSP245 and SSP585 scenarios from 2015 to 2100 (Fig. S20). Notably, we observe a pronounced south-north difference, particularly under the SSP585 scenario, indicating that the non-uniform enhancement of T_w due to global warming is reasonably robust. Regional downscaling covering longer decades would contribute to better

projections for China if computational resources permit.”

Fig. R2.26. Future variations of T_w . Spatial distribution of T_w variations under the SSP2-4.5 (A) and SSP5-8.5 (B) scenarios during the period of 2015 to 2100. Black dots denote areas with significant trend ($P < 0.05$).

Fig. S2: Clarify the method of obtaining T and E_a sensitivity. Eliminate unrealistic configurations where E_a exceeds saturated water vapor pressure (E_s) from the figure to avoid misleading.

Reply:

- Thanks for your reminder. Based on Equation (1)-(3), we calculated T_w with T and E_a ranging from 10°C to 40°C and 10 hPa to 40 hPa, respectively. If E_a exceeded saturated water vapor pressure, RH was set to 100%. We have clarified that in Materials and Methods and modified Fig. S2 (Fig. R2.27 here).

$$RH = \frac{E_a}{E_s} \times 100\% \quad (1)$$

$$E_s = 6.112 \times e^{\frac{17.62 \times T}{T + 243.12}} \quad (2)$$

$$T_w = T \cdot \operatorname{atan} \left[0.151977(RH + 8.313659)^{\frac{1}{2}} \right] + \operatorname{atan}(T + RH) - \operatorname{atan}(RH - 1.676331) + 0.00391838(RH)^{\frac{3}{2}} \cdot \operatorname{atan}(0.023101 \cdot RH) - 4.686035 \quad (3)$$

- “Using Equation (1)-(3), we obtained T and E_a sensitivity by calculating T_w with T and E_a ranging from 10°C to 40°C and 10 hPa to 40 hPa, respectively. In cases where E_a exceeded the saturated water vapor pressure, RH was set to 100%.”

Fig. R2.27. Sensitivity of T_w to air temperature (T) and E_a . The dashed black line indicates saturated water vapor pressure.

Fig. S5: Provide technical details on how the Probability Density Functions (PDFs) were derived. How did the authors exclude the impacts other than land use change? Explain how the urban and rural stations are identified in the text.

Reply:

- Sorry for the vagueness.
- Those sites where E_a has significant trends from 1979 to 2018 were divided into two categories, with and without land use conversion. For each category, the PDFs were computed based on normal distribution by clustering the trends within several equal-length bins and using the following equation,

$$f(x) = \frac{1}{\sigma\sqrt{2\pi}} e^{-\frac{(x-\mu)^2}{2\sigma^2}}$$

where σ is the sample standard deviation and μ is the mean value.

- We didn't exclude the impacts other than land use change. The PDF of E_a changes indicates that E_a is more likely to decrease in areas with land use conversion, which may contribute to the decrease or slower increase of T_w . Our attribution analysis considering single forcing of land use change further verifies the role of land use change on T_w variations.
- We used gridded land use dataset obtained from National Tibetan Plateau Data Center (<https://data.tpdac.ac.cn>) to identify the time of the land use conversion. The data set included seven periods: the end of 1980s, 1990, 1995, 2000, 2005, 2010 and 2015. We determine conversion time when the first time the station location

grid changed to urban land. For example, if land use type of one station is nonurban in 1995 but becomes urban land use in 2000, we determine the conversion time of this station to be 2000.

- To avoid confusion, we added related expressions in the revised manuscript.
- “We determined conversion time when the first time the station location grid changed to urban land use. The PDFs for changes of E_a in areas with and without land use conversion were computed based on normal distribution using the following equation,

$$f(x) = \frac{1}{\sigma\sqrt{2\pi}} e^{-\frac{(x-\mu)^2}{2\sigma^2}}$$

where σ is the sample standard deviation and μ is the mean value.”

Figs. S6-8: Do the authors refer to “anthropogenic” aerosol experiment?

Reply:

- Thank you for pointing out this. It should be anthropogenic aerosol. We have clarified this in both Results, and Methods and Materials”
- “We find the dominant influence of greenhouse gases (GHG) on variations of T_w (Fig. S14, approximately 63%), followed by anthropogenic aerosols (AER, ~ 24%) (Fig. S14).”
- “The historical simulations of monthly T and RH driven by time evolving all forcings and single forcing from the Detection and Attribution Model Intercomparison Project (DAMIP) 76 and the Land Use Model Intercomparison Project (LUMIP) 77 in the Coupled Model Intercomparison Project Phase 6 (CMIP6) were employed in this study to assess climate response to individual forcings, including GHG-only (GHG), anthropogenic aerosols-only (AER), natural-only (NAT) and land use-only (LU).”

References **and** **dataset**
Argiriou A A, Li Z, Armaos V, Mamara A, Shi Y L, Yan Z W. 2023: Homogenised Monthly and Daily Temperature and Precipitation Time Series in China and Greece since 1960. Adv. Atmos. Sci., doi: 10.1007/s00376-022-2246-4

Li, Z., Z. W. Yan, Y. N. Zhu, N. Freychet, and S. Tett, 2020: Homogenized daily relative humidity series in China during 1960–2017. Adv. Atmos. Sci., 37(4), 318–327, <https://doi.org/10.1007/s00376-020-9180-0>

Daily temperature and relative humidity data link
<https://doi.org/10.57760/sciencedb.01731>
<https://doi.org/10.11922/sciencedb.804>

Ding et al. (2008) Inter-decadal variation of the summer precipitation in East China and

its association with decreasing Asian summer monsoon. Part I: Observed evidences. Int. J. Climatol. 28, 1139–1161.

Ding et al. (2009) *Interdecadal variation of the summer precipitation in China and its association with decreasing Asian summer monsoon. Part II: Possible causes. Int. J. Climatol.*, 29, 1926–1944, doi:10.1002/joc.1759

Huang et al. (2023) *Relative contributions of internal variability and external forcing to the inter-decadal transition of climate patterns in East Asia. npj Clim Atmos Sci* 6, 21. <https://doi.org/10.1038/s41612-023-00351-0>

Zhang, Y. et al. (2021) *Projections of tropical heat stress constrained by atmospheric dynamics. Nat. Geosci.* 14, 133–137. <https://doi.org/10.1038/s41561-021-00695-3>

REVIEWER COMMENTS

Reviewer #1 (Remarks to the Author):

Thanks for the authors' responses and most issues arised before have been satisfactorily addressed and corrected. For the response of my Q6, the authors mentioned that the EMD and EOF methods are used for analysis of the detection and contribution of GHG, AER, NAT to the change in Tw. I still wonder these methods can resolve this issue. In general, the detection and attribution techneque of the optimal fingerprinting method is used.

Reviewer #2 (Remarks to the Author):

Comments on the manuscript entitled "Widely-spread and uniformly elevated future heat stress in China driven by spatially heterogeneous response of water vapor" by Wang et al submitted to Nature Communications.

The authors have properly addressed most of my previous comments; however, there are still outstanding issues that need to be resolved before considering the manuscript for publication in Nature Communications. Below are detailed comments:

In Lines 111-118, the authors stated that "T dominates still changes of Tw in northern China..." and "Ea trends exhibit a distinct North-South divide, characterized by higher enhancement over northern stations...". These statements seem contradictory. If Ea increases faster in northern stations compared to southern ones, why does T, rather than Ea, dominate the change in Tw in northern China? It would be beneficial to provide a "quantitative" comparison here regarding the relative contribution of T and Ea to Tw.

The mechanism analysis employed correlation and composite analysis method (e.g., Lines 153-156). It should be noted that while a significant correlation is observed, it may not necessarily imply direct physical connections, especially when the time series exhibit clear trends. It appears that the correlation between Tw and SAH/WPSH is largely influenced by their long-term trends. I suggest that the authors remove the long-term trend of Tw and SAH/WPSH and re-calculate the correlation. Demonstrating a statistically significant correlation coefficient even after removing the long-term trend would increase confidence in the mechanism analysis. Additionally, the authors claimed changes in SAH and WPSH lead to descending motion (Lines 150-153); however, relevant figures supporting this claim are missing.

The authors mentioned in Lines 164-165 that "We find the dominant influence of greenhouse gases on variations of Tw". Please clarify the specific regions where GHGs dominate.

Line 191: replace "form" with "from"

Line 201: Please consider revising “Bay of Bangladesh” to “Bay of Bengal” and “lower layers” to “lower troposphere”

Lines 200-202: Please elaborate on how the intensification of SAH and WPSH leads to an anticyclonic anomaly over the Bay of Bengal and South China Sea.

In Lines 326-328 and Eq. (4), the authors calculated the linear regression of various external forcings onto T_w using MME data, which largely removes internal climate variability. As the authors only utilized 36-year data from 1979 to 2014 in this study, the low-frequency variability such as IPO and AMO may also affect the long-term trend of T_w . For example the rapid warming after the late-1970s is partly attributed to internal climate variability, e.g., IPO. Therefore, the long-term trend derived from EMD may include internal climate variability in addition to external forcing. This caveat should be discussed thoroughly.

Please correct “SSP245 and SSP585” to “SSP2-4.5 and SSP5-8.5” in Line 344.

In Line 354, please specify which FDDA method was used, spectral nudging or grid nudging. Additionally, clarify the nudging coefficient and wavelength if spectral nudging was employed.

Line 598: “Green dashed line” or “Yellow dashed line”?

Response to Reviewers

Reviewer #1 (Remarks to the Author):

Thanks for the authors' responses and most issues arised before have been satisfactorily addressed and corrected. For the response of my Q6, the authors mentioned that the EMD and EOF methods are used for analysis of the detection and contribution of GHG, AER, NAT to the change in T_w . I still wonder these methods can resolve this issue. In general, the detection and attribution techneque of the optimal fingerprinting method is used.

Reply:

- We appreciate your recognition of our revised manuscript. In the manuscript, we already used a total least-squares optimal fingerprinting approach to attribute the contribution of each single forcing to variations of T_w under all forcing conditions. Additionally, to further evaluate contributions of external forcings to changes in T_w from observations, we conducted EMD analysis to separate internal natural variability and external anthropogenic forcing. After it, we employed EOF decomposition on derived T_w values resulting from external forcings to identify the contributions of each anthropogenic factor including GHG, AER and LU. The results from EMD and EOF are generally consistent with the results from the total least-squares optimal fingerprinting approach.
- To further address your concern about the EMD and EOF methods on detecting the contribution of GHG, AER, NAT and LU to the changes in T_w , we added more explanations in the revised manuscript. EMD is a data-adaptive multiresolution technique to decompose a signal into physically meaningful components. It can be used to analyze non-linear and non-stationary signals by separating them into components at different resolutions (Huang et al., 1998). Decomposed signals can be considered as periodic oscillations having different frequencies (Lee and Ouarda, 2011; Wu et al., 2007), while the residual obtained from the EMD analysis can be considered as the impacts of anthropogenic emissions, urbanization and land use change (Loehle and Scafetta, 2011).
- EOF analysis on external signals can extract the dominant patterns or modes of variability, which can linked to the contributions of each anthropogenic factor including GHG, AER and LU. EOF1 and EOF2 can be considered as the contributions of GHG and AER, respectively, as they exhibit similar spatial patterns as our attribution analysis (Fig. R1.1A, B, Fig. R1.2C, D). They account for 68.6% and 23.8% of the total variations, respectively, which closely aligns with the results obtained from our attribution analysis (GHG: ~63%, AER: ~24%). The third mode of EOF shows a generally decreasing trend of T_w , which can be attributed to the impacts of urbanization and land use changes (Fig. R1.1C).
- We added these explanations in the revised manuscript.

- “We also employed EMD decomposition on T_w variations to separate internal natural variability and external anthropogenic forcing. EMD is a data-adaptive multiresolution technique to decompose non-linear and non-stationary signals by separating them into physically meaningful components at different resolutions ⁷⁷. Decomposed signals can be considered as periodic oscillations with different frequencies ^{78,79}, while the residual can capture impacts of external forcings ⁸⁰. Additionally, we conducted EOF analysis on external signals to obtain the dominant patterns or modes of variability and linked them to the contributions of each anthropogenic factor including GHG, AER and LU.”

Fig. R1.1. Empirical orthogonal function (EOF) decomposition of T_w . Spatial patterns of (A) EOF1, (B) EOF2, and (C) EOF3.

Fig. R1.2. Observed and simulated variations of T_w . (A) Spatial distribution of observed T_w variations during the period of 1979-2014. Only sites with significant trend ($P < 0.05$) are displayed. (B) Spatial distribution of simulated T_w variations under all-forcing conditions during the period of 1979-2014. Black dots denote areas with significant trend ($P < 0.05$). Spatial distribution of simulated T_w variations under GHG-only (C), aerosols-only (D), natural-only (E) and land use-only (F) forcing conditions during the period of 1979-2014. Black dots denote areas with significant trend ($P < 0.05$).

Reference:

- Huang, N.E., Shen, Z., Long, S.R., Wu, M.C., Shih, H.H., Zheng, Q., Yen, N.C., Tung, C.C. and Liu, H.H., 1998. The empirical mode decomposition and the Hilbert spectrum for nonlinear and non-stationary time series analysis. *Proceedings of the Royal Society of London. Series A: mathematical, physical and engineering sciences*, 454(1971), pp.903-995.

- Loehle, C. and Scafetta, N., 2012. Climate change attribution using empirical decomposition of climatic data. arXiv preprint arXiv:1206.5845.
- Lee, T. and Ouarda, T.B., 2011. Prediction of climate nonstationary oscillation processes with empirical mode decomposition. *Journal of Geophysical Research: Atmospheres*, 116(D6).
- Wu, Z., Huang, N.E., Long, S.R. and Peng, C.K., 2007. On the trend, detrending, and variability of nonlinear and nonstationary time series. *Proceedings of the National Academy of Sciences*, 104(38), pp.14889-14894.

Reviewer #2 (Remarks to the Author):

Comments on the manuscript entitled “Widely-spread and uniformly elevated future heat stress in China driven by spatially heterogeneous response of water vapor” by Wang et al submitted to Nature Communications.

The authors have properly addressed most of my previous comments; however, there are still outstanding issues that need to be resolved before considering the manuscript for publication in Nature Communications. Below are detailed comments:

Reply:

- Thank you for your time to evaluate our revised manuscript. In the revised version of our manuscript, we have carefully read and addressed all remaining concerns. Specific responses and revisions to each comment are provided in detail below.

In Lines 111-118, the authors stated that “T dominates still changes of T_w in northern China...” and “ E_a trends exhibit a distinct North-South divide, characterized by higher enhancement over northern stations...”. These statements seem contradictory. If E_a increases faster in northern stations compared to southern ones, why does T, rather than E_a , dominate the change in T_w in northern China? It would be beneficial to provide a “quantitative” comparison here regarding the relative contribution of T and E_a to T_w .

Reply:

- Thank you for this valuable comment. Both E_a and T increase faster in northern China (Fig. R2.1A), and both of them are important for changes in T_w in northern China. While E_a mostly shows decreasing trend in southern China (Fig. R2.1B), decreases in T_w in southern China are dominated by E_a changes. Difference between changes in T_w in southern and northern China is caused by varied E_a changes across China. We agree that the original expression on this is vague.
- Following your suggestion of providing a “quantitative” comparison regarding the relative contribution of T and E_a to T_w , we calculated relative contributions of E_a and T at each site by regressing observed T_w variation with changes in T_w when either T or E_a varied but the other factor fixed at year 1979 (Fig. R2.1). Given that T generally rises, whereas E_a and T_w exhibit both increasing and decreasing trends throughout China, we observe a wide range of positive and negative percentage contributions of T and E_a to changes in T_w (Fig. R2.1E, F). A greater positive percentage contribution from either E_a or T can be interpreted

as playing a dominant role in influencing T_w variations (Fig. R2.1G).

- We found that T_w changes in southern China are dominated by E_a changes, while T dominates changes of T_w at more sites in northern China although E_a changes are also important in some regions of northern China. To avoid confusion, we added quantitative comparison in the revised Fig. S4 (Fig. R2.1 here) and revised related descriptions in the manuscript.
- “Changes in T_w when either T or E_a varied but the other factor fixed at year 1979 highlight that E_a changes are responsible for the observed different T_w trends in southern China (Fig. 2 and Fig. S4).”

Fig. R2.1. Air temperature (T), water vapor (E_a) and induced T_w variations. (A) Spatial distribution of T variations during the period from 1979 to 2018. Only sites with significant trend ($P < 0.05$) are displayed. (B) Spatial distribution of E_a variations during

the period from 1979 to 2018. Only sites with significant trend ($P < 0.05$) are displayed. (C) Spatial distribution of T induced T_w variations during the period from 1979 to 2018. Only sites with significant trend ($P < 0.05$) are displayed. (D) Spatial distribution of E_a induced T_w variations during the period from 1979 to 2018. Only sites with significant trend ($P < 0.05$) are displayed. Percentage contribution of T_w induced by T (E) and E_a (F) to total T_w changes during the period of 1979-2018. (G) Dominant role of T and E_a on T_w changes. Red indicates the dominant role of T, while blue indicates the dominant role of E_a .

The mechanism analysis employed correlation and composite analysis method (e.g., Lines 153-156). It should be noted that while a significant correlation is observed, it may not necessarily imply direct physical connections, especially when the time series exhibit clear trends. It appears that the correlation between T_w and SAH/WPSH is largely influenced by their long-term trends. I suggest that the authors remove the long-term trend of T_w and SAH/WPSH and re-calculate the correlation. Demonstrating a statistically significant correlation coefficient even after removing the long-term trend would increase confidence in the mechanism analysis. Additionally, the authors claimed changes in SAH and WPSH lead to descending motion (Lines 150-153); however, relevant figures supporting this claim are missing.

Reply:

- Thank you for your valuable suggestions. We re-calculate the correlation between T_w and SAH/WPSH after removing the long-term trend (Fig. R2.2). We can still find statistically significant (at 95% level) correlation coefficients at most sites, highlighting the connection between T_w and SAH/WPSH. We replaced Fig. S10 with Fig. R2.2 in the revised manuscript.
- Changes in SAH and WPSH lead to northward movement of high-pressure center at upper levels, making South China at edge regions (Fig. R2.3A, B), where descending motion prevails (Fig. R2.3E, F). We added subgraphs showing vertical velocities in Fig. S8 (Fig. R2.3 here) into the revised manuscript to improve readability.

Fig. R2.2. Correlations of the SAH and WPSH with variations of T_w . Correlations of the detrended latitudes of the SAH eastward ridge point (ERP) (A) and WPSH westward ridge point (WRP) (B) with variations of detrended T_w . Correlations of the detrended longitudes of the SAH eastward ridge point (ERP) (C) and WPSH westward ridge point (WRP) (D) with variations of detrended T_w . Only sites having significant trend with 95% and higher confidence level are displayed.

Fig. R2.3. Regression of atmospheric features on the first leading mode. Regression of geopotential height and circulation at 100 hPa (A), 500 hPa (B), 700 hPa (C) and 850 hPa (D) on the first leading mode. Red dots denote areas with significant correlation ($P < 0.05$). Blue solid lines in A and B indicate climatologically averaged locations of the SAH (represented by 16760-dagpm line) and the WPSH (represented by 5880-dagpm line), respectively. Blue dashed lines in A and B indicate varied locations of the SAH and WPSH, respectively. Regression of vertical velocity at 700 hPa (E) and 850 hPa (F) on the first leading mode. Positive values indicate descending motion. Black dots denote areas with significant correlation ($P < 0.05$).

The authors mentioned in Lines 164-165 that “We find the dominant influence of greenhouse gases on variations of T_w ”. Please clarify the specific regions where GHGs dominate.

Reply:

- Following your comments, we revised Fig. S16 to the percentage contributions of individual forcing to variations of T_w . We can find dominant influence of greenhouse gases on variations of T_w across China except the northeastern desert regions (Fig. R2.4). The dominant influence of greenhouse gases is more pronounced over South China (Fig. R2.4A). We revised related expressions in the manuscript accordingly.
- “We find the dominant influence of greenhouse gases (GHG) on variations of T_w (Fig. S15, Fig. S16A, approximately 63%), followed by aerosols (AER, ~24%) (Fig. S15, Fig. S16B). GHG dominates T_w changes across China, except in the northeastern desert regions (Fig. S16E), with a more pronounced effect over South China (Fig. S16A)”

Fig. R2.4. Percentage contributions of individual forcing to variations of T_w .

Percentage contributions of T_w from GHG-only (A), aerosols-only (B), natural-only (C) and land use-only (D) forcing conditions to total T_w variations during the period of 1979-2014. (E) Dominant role of GHG and aerosols on T_w changes. Red indicates the dominant role of GHG, while yellow indicates the dominant role of aerosols.

Line 191: replace “form” with “from”

Reply:

- Done as suggested.

Line 201: Please consider revising “Bay of Bangladesh” to “Bay of Bengal” and “lower layers” to “lower troposphere”

Reply:

- Done as suggested.

Lines 200-202: Please elaborate on how the intensification of SAH and WPSH leads to an anticyclonic anomaly over the Bay of Bengal and South China Sea.

Reply:

- Thank you for your time to offer valuable suggestions. Intensification of the SAH and WPSH results in the zonal nearing of these two systems (Chen and Zhai, 2016), leading to weakened tropical easterly jet (Li et al., 2021) but accelerated westerly winds over subtropical regions at upper levels (Zhang et al., 2024; Wang, 2020) (Fig. R2.5B, D). This process may enhance the eastward propagation of Kelvin waves (Xue et al., 2018; Xie et al., 2009), intensifying the convergence of cross-equatorial flows (CEFs) (Hastenrath, 2002; Sun et al., 2022). The increased convergence of CEFs triggers descending motion over tropical areas (Tomas and Webster, 1997), resulting in anticyclonic anomalies over the Bay of Bengal and South China Sea at lower levels (Fig. R2.5E, F, G, H). This atmospheric circulation anomaly is similar to the one associated with the extreme dry-wet contrast event between southern and northern China in the summer of 2020 (Wang et al., 2024; Du et al., 2022), confirming the influence of such atmospheric structures on future projected faster increase in E_a in northern China. We added this explanation in the revised manuscript.
- “Under a warming climate, both SAH and WPSH exhibit substantial intensification, with a notable increase in central pressure (Fig. S21). This leads to zonal nearing of these two systems and weakened tropical easterly jet⁵¹ but accelerated westerly winds over subtropical regions at upper troposphere^{52,53}

(Fig. S23C, D). The accelerated westerly wind enhances the eastward propagation of Kelvin waves^{54,55} and convergence of cross-equatorial flows^{56,57}. As a result, the descending motion is intensified over tropical regions⁵⁸ and consequently anomalous anticyclone is found over the Bay of Bengal and South China Sea at lower troposphere (Fig. S23E, F, G, H). Similar atmospheric circulation anomaly has been observed during the extreme dry-wet contrast event between southern and northern China in 2020^{59,60},

Fig. R2.5. Future changes of atmospheric circulations. Spatial distribution of future changes of summertime geopotential height and circulation at 100 hPa under the SSP2-4.5 (A) and SSP5-8.5 (B) scenarios from bias corrected CMIP6 global dataset. Spatial

distribution of future changes of summertime geopotential height and circulation at 500 hPa under the SSP2-4.5 (C) and SSP5-8.5 (D) scenarios from bias corrected CMIP6 global dataset. Spatial distribution of future changes of summertime geopotential height and circulation at 700 hPa under the SSP2-4.5 (E) and SSP5-8.5 (F) scenarios from bias corrected CMIP6 global dataset. Spatial distribution of future changes of summertime geopotential height and circulation at 850 hPa under the SSP2-4.5 (G) and SSP5-8.5 (H) scenarios from bias corrected CMIP6 global dataset.

Reference:

- Hastenrath, S., 2002. The intertropical convergence zone of the eastern Pacific revisited. *International Journal of Climatology: A Journal of the Royal Meteorological Society*, 22(3), pp.347-356.
- Sun, C., Liu, Y., Wei, T., Kucharski, F., Li, J. and Wang, C., 2022. Cross-hemispheric SST propagation enhances the predictability of tropical western Pacific climate. *npj Climate and Atmospheric Science*, 5(1), p.38.
- Tomas, R.A. and Webster, P.J., 1997. The role of inertial instability in determining the location and strength of near-equatorial convection. *Quarterly Journal of the Royal Meteorological Society*, 123(542), pp.1445-1482.
- Wang, J., 2020. Relationships between Jianghuai Meiyu anomaly and the collaborative evolution of wave trains in the upper and lower troposphere in mid-July of 2020. *Frontiers in Earth Science*, 8, p.597930.
- Zhang, J., Yue, P., Zhao, J. and Yang, Y., 2024. Dipolar mode of summer precipitation over the Upper Yellow River Basin in China and possible causes. *Theoretical and Applied Climatology*, pp.1-13.
- Chen, Y. and Zhai, P., 2016. Mechanisms for concurrent low-latitude circulation anomalies responsible for persistent extreme precipitation in the Yangtze River Valley. *Climate Dynamics*, 47, pp.989-1006.
- Xue, F., Dong, X. and Fan, F., 2018. Anomalous western Pacific subtropical high during El Niño developing summer in comparison with decaying summer. *Advances in Atmospheric Sciences*, 35, pp.360-367.
- Xie, S.P., Hu, K., Hafner, J., Tokinaga, H., Du, Y., Huang, G. and Sampe, T., 2009. Indian Ocean capacitor effect on Indo-western Pacific climate during the summer following El Niño. *Journal of climate*, 22(3), pp.730-747.
- Li, Z., Sun, Y., Li, T., Chen, W. and Ding, Y., 2021. Projections of South Asian summer monsoon under global warming from 1.5 to 5 C. *Journal of Climate*, 34(19), pp.7913-7926.
- Du, J., Fu, K., Wang, K. and Cui, B., 2022. Anthropogenic influences on 2020 extreme

dry-wet contrast over South China. Bull. Amer. Meteor. Soc., 103(3), pp.S68-S75.

- Wang, K., Zheng, Z., Zhu, X., Dong, W., Tett, S.F., Dong, B., Zhang, W., Lott, F.C., Bu, L., Wang, Y. and Li, H., Anthropogenic Influences on the Extremely Dry and Hot Summer of 2020 in Southern China and Projected Changes in the Likelihood of the Event. Available at SSRN 4773598.

In Lines 326-328 and Eq. (4), the authors calculated the linear regression of various external forcings onto T_w using MME data, which largely removes internal climate variability. As the authors only utilized 36-year data from 1979 to 2014 in this study, the low-frequency variability such as IPO and AMO may also affect the long-term trend of T_w . For example the rapid warming after the late-1970s is partly attributed to internal climate variability, e.g., IPO. Therefore, the long-term trend derived from EMD may include internal climate variability in addition to external forcing. This caveat should be discussed thoroughly.

Reply:

- We acknowledge that the low-frequency variability such as IPO and AMO may also affect the long-term trend of T_w (Zhang et al., 2020), and MME can largely remove internal climate variability (Jiang et al., 2023). For original observations, we employed empirical decomposition method (EMD) to separate internal natural variability and external anthropogenic forcing. The residual obtained from the EMD can be considered as the impacts of anthropogenic emissions, urbanization and land use change (Loehle and Scafetta, 2011). We found the internal variability caused insignificant trends in T_w at most sites of China (Fig. R2.6A). However, external forcings emerge as the dominant factor driving T_w changes during the period of 1979-2018 (Fig. R2.6B). This is evident from the consistent spatial patterns observed, which align with our original findings. We further employed empirical orthogonal function (EOF) decomposition on derived T_w values resulting from external forcings to identify the contributions of each anthropogenic factor including GHG, AER and LU (Fig. R2.7). EOF1 and EOF2 can be considered as the contributions of GHG and AER, respectively, as they account for 68.6% and 23.8% of the total variations (Fig. R2.7A, B), respectively, which closely aligns with the results obtained from our attribution analysis (GHG: ~63%, AER: ~24%). The third mode of EOF shows a generally decreasing trend of T_w , which can be attributed to the impacts of urbanization and land use changes (Fig. R2.7C).
- This means internal climate variability has relatively small impact on T_w variations which will not affect main findings of our study. We considered this as a limitation of this study and discussed it in the revised manuscript.
- “While multi-model ensembles (MME) data remove internal climate variability

signals, the influence of low-frequency variability⁵⁹, such as the Interdecadal Pacific Oscillation (IPO) and the Atlantic Multidecadal Oscillation (AMO), could still impact the long-term trend of T_w ⁶⁰. The alignment of EMD analysis (Fig. S17) and EOF decomposition (Fig. S18) on observed T_w variations with the results obtained from attribution analysis (Fig. S15) demonstrates that the impact of internal climate variability on T_w variations during the study period is relatively small, which will not affect our major findings here.”

- EMD is a data-adaptive multiresolution technique to decompose non-linear and non-stationary signals by separating them into physically meaningful components at different resolutions (Huang et al., 1998). Decomposed signals obtained from the EMD analysis can be considered as periodic oscillations having different frequencies (Lee and Ouarda, 2011; Wu et al., 2007), while the residual can be considered as the impacts of external forcings (Loehle and Scafetta, 2011). By subtracting the residual obtained from the EMD analysis in the observed T_w , we can obtain T_w variations caused by internal climate variability. We added this explanation in the revised manuscript.
- “We also employed EMD decomposition on T_w variations to separate internal natural variability and external anthropogenic forcing. EMD is a data-adaptive multiresolution technique to decompose non-linear and non-stationary signals by separating them into physically meaningful components at different resolutions⁷⁷. Decomposed signals can be considered as periodic oscillations with different frequencies^{78,79}, while the residual can capture impacts of external forcings⁸⁰.”

Fig. R2.6. Empirical decomposition (EMD) analysis on T_w . Spatial distribution of T_w trends caused by natural variabilities (A) and external forcings (B) during the period of 1979-2018.

Fig. R2.7. Empirical orthogonal function (EOF) decomposition of T_w . Spatial patterns of (A) EOF1, (B) EOF2, and (C) EOF3.

Reference:

- Huang, N.E., Shen, Z., Long, S.R., Wu, M.C., Shih, H.H., Zheng, Q., Yen, N.C., Tung, C.C. and Liu, H.H., 1998. The empirical mode decomposition and the Hilbert spectrum for nonlinear and non-stationary time series analysis. *Proceedings of the Royal Society of London. Series A: mathematical, physical and engineering sciences*, 454(1971), pp.903-995.
- Loehle, C. and Scafetta, N., 2012. Climate change attribution using empirical decomposition of climatic data. *arXiv preprint arXiv:1206.5845*.
- Lee, T. and Ouarda, T.B., 2011. Prediction of climate nonstationary oscillation processes with empirical mode decomposition. *Journal of Geophysical Research: Atmospheres*, 116(D6).
- Wu, Z., Huang, N.E., Long, S.R. and Peng, C.K., 2007. On the trend, detrending, and variability of nonlinear and nonstationary time series. *Proceedings of the National Academy of Sciences*, 104(38), pp.14889-14894.
- Zhang, G., Zeng, G., Li, C. and Yang, X., 2020. Impact of PDO and AMO on interdecadal variability in extreme high temperatures in North China over the most recent 40-year period. *Climate Dynamics*, 54(5), pp.3003-3020.

Please correct “SSP245 and SSP585” to “SSP2-4.5 and SSP5-8.5” in Line 344.

Reply:

- Done as suggested.

In Line 354, please specify which FDDA method was used, spectral nudging or grid nudging. Additionally, clarify the nudging coefficient and wavelength if spectral nudging was employed.

Reply:

- We used grid nudging and the used nudging coefficients for winds, air temperature and water vapor were all 0.0003.
- We have added this detailed information in the revised manuscript.
- “To enhance the accuracy of simulated meteorological variables, the four-dimensional data assimilation (FDDA) technique was adopted, effectively nudging horizontal winds, temperature, and moisture across all vertical levels. We used grid nudging and the adopted nudging coefficients for these variables were all 0.0003.”

Line 598: “Green dashed line” or “Yellow dashed line”?

Reply:

- We revised lines in Fig. 1 and 2 to black dashed lines to enhance readability.

Fig. 1. Spatiotemporal variations of wet bulb temperature (T_w) in China. (A) Spatial distribution of average summertime T_w during the period from 1979 to 2018. Black squares represent four key agglomerations: Beijing-Tianjin-Hebei (BTH, 38°N-41°N, 115°E-120°E), Yangtze River Delta (YRD, 29°N-33°N, 118°E-123°E), Sichuan Basin (SCB, 29°N-32°N, 103°E-107°E) and Pearl River Delta (PRD, 21°N-23°N, 112°E-115°E). (B) Spatial distribution of T_w trend during the period from 1979 to 2018. Only sites with significant trend ($P < 0.05$) are displayed. Black dashed line indicates the latitude of 33°N. (C) Time series of average T_w anomaly of northern and southern

stations during the period from 1979 to 2018. (D) Time series of average T_w of BTH, YRD, SCB and PRD during the period from 1979 to 2018.

Fig. 2. Air temperature (T) and water vapor (E_a) variations. (A) Spatial distribution of T variations during the period from 1979 to 2018. Only sites with significant trend ($P < 0.05$) are displayed. Black dashed line indicates the latitude of 33°N . (B) Time series of average T anomaly of northern and southern stations during the period from 1979 to 2018. (C) Spatial distribution of E_a variations during the period from 1979 to 2018. Only sites with significant trend ($P < 0.05$) are displayed. Black dashed line indicates 33°N . (D) Time series of average E_a anomaly of northern and southern stations during the period from 1979 to 2018.

REVIEWERS' COMMENTS

Reviewer #2 (Remarks to the Author):

The authors presented an interesting study on the historical changes and future projections of wet bulb temperature over China. All of my comments were reasonably addressed. Therefore, I recommend accepting the manuscript.